# The covariance perceptron: A new paradigm for classification and processing of time series in recurrent neuronal networks

Matthieu Gilson[1,2☯]*, David Dahmen[2☯], Rubén Moreno-Bote[1,3], Andrea Insabato[4], Moritz Helias[2,5]

**1** Center for Brain and Cognition, Department of Information and Telecommunication technologies, Universitat Pompeu Fabra, Barcelona, Spain, **2** Institute of Neuroscience and Medicine (INM-6) and Institute for Advanced Simulation (IAS-6) and JARA Institute Brain Structure-Function Relationships (INM-10), Jülich Research Centre, Jülich, Germany, **3** ICREA, Barcelona, Spain, **4** IDIBAPS (Institut d'Investigacions Biomèdiques August Pi i Sunyer), Barcelona, Spain, **5** Department of Physics, Faculty 1, RWTH Aachen University, Aachen, Germany

☯ These authors contributed equally to this work.
\* matthieu.gilson@upf.edu

**Data Availability Statement:** Example Python scripts with implementations of the learning rules to reproduce some key figures are available at https://github.com/MatthieuGilson/covariance_

## Abstract

Learning in neuronal networks has developed in many directions, in particular to reproduce cognitive tasks like image recognition and speech processing. Implementations have been inspired by stereotypical neuronal responses like tuning curves in the visual system, where, for example, ON/OFF cells fire or not depending on the contrast in their receptive fields. Classical models of neuronal networks therefore map a set of input signals to a set of activity levels in the output of the network. Each category of inputs is thereby predominantly characterized by its mean. In the case of time series, fluctuations around this mean constitute noise in this view. For this paradigm, the high variability exhibited by the cortical activity may thus imply limitations or constraints, which have been discussed for many years. For example, the need for averaging neuronal activity over long periods or large groups of cells to assess a robust mean and to diminish the effect of noise correlations. To reconcile robust computations with variable neuronal activity, we here propose a conceptual change of perspective by employing variability of activity as the basis for stimulus-related information to be learned by neurons, rather than merely being the noise that corrupts the mean signal. In this new paradigm both afferent and recurrent weights in a network are tuned to shape the input-output mapping for covariances, the second-order statistics of the fluctuating activity. When including time lags, covariance patterns define a natural metric for time series that capture their propagating nature. We develop the theory for classification of time series based on their spatio-temporal covariances, which reflect dynamical properties. We demonstrate that recurrent connectivity is able to transform information contained in the temporal structure of the signal into spatial covariances. Finally, we use the MNIST database to show how the covariance perceptron can capture specific second-order statistical patterns generated by moving digits.

perceptron. The original MNIST dataset underlying the results presented in the study are available from http://yann.lecun.com/exdb/mnist/. The remainder of the results rely on synthetic data and algorithms that can be implemented following the information in the manuscript.

**Funding:** This work was partially supported by the European Union's Horizon 2020 research and innovation programme under grant agreement No. 785907 (Human Brain Project SGA2). MG acknowledges funding from the Marie Sklodowska-Curie Action (Grant H2020-MSCA-656547) of the European Commission. DD and MH acknowledge the Helmholtz young investigator's group (VH-NG-1028), the Exploratory Research Space (ERS) seed fund neuroIC002 (EXS-SF-neuroIC002) of the RWTH university and the JARA Center for Doctoral studies within the graduate School for Simulation and Data Science (SSD). AI acknowledges funding from the Marie Sklodowska-Curie Action (Grant H2020-MSCA-841684) of the European Commission. RMB acknowledges funding from the Howard Hughes Medical Institute (HHMI, ref 55008742), MINECO (Spain; BFU2017-85936-P) and ICREA Academia (2016). The funders had no role in study design, data collection and analysis, decision to publish, or preparation of the manuscript.

**Competing interests:** The authors have declared that no competing interests exist.

## Author summary

The dynamics in cortex is characterized by highly fluctuating activity: Even under the very same experimental conditions the activity typically does not reproduce on the level of individual spikes. Given this variability, how then does the brain realize its quasi-deterministic function? One obvious solution is to compute averages over many cells, assuming that the mean activity, or rate, is actually the decisive signal. Variability across trials of an experiment is thus considered noise. We here explore the opposite view: Can fluctuations be used to actually represent information? And if yes, is there a benefit over a representation using the mean rate? We find that a fluctuation-based scheme is not only powerful in distinguishing signals into several classes, but also that networks can efficiently be trained in the new paradigm. Moreover, we argue why such a scheme of representation is more consistent with known forms of synaptic plasticity than rate-based network dynamics.

## Introduction

A fundamental cognitive task that is commonly performed by humans and animals is the classification of time-dependent signals. For example, in the perception of auditory signals, the listener needs to distinguish the meaning of different sounds: The neuronal system receives a series of pressure values, the stimulus, and needs to assign a category, for example whether the sound indicates the presence of a predator or a prey. Neuronal information processing systems are set apart from traditional paradigms of information processing by their ability to be trained, rather than being algorithmically programmed. The same architecture, a network composed of neurons connected by synapses, can be adapted to perform different classification tasks. The physical implementation of learning predominantly consists of adapting the connection strengths between neurons —a mechanism termed synaptic plasticity. Earlier models of plasticity like the Hebbian rule [1, 2] focused on the notion of firing together, which was interpreted in terms of firing rate. In parallel to such unsupervised learning rules, supervised learning and reinforcement learning have also been explored to explain how biological systems can be trained to perform cognitive tasks, such as pattern recognition [3–5].

The representation of the stimulus identity by the mean firing activity alone is, however, challenged by two observations in biological neuronal networks. First, synaptic plasticity, the biophysical implementation of learning, has been shown to depend on the relative temporal spiking activity of the presynaptic and the postsynaptic neurons [6, 7], which can be formalized in terms of the covariance of the neuronal activity [8, 9]. Examples of second-order statistics of the spiking activity that induce strong weight specialization not only include the canonical example of spike patterns with reliable latencies, like spike volleys following visual stimulation [10], but also a great variety of spiking statistics such as fast stereotypical co-fluctuations even in the case of Poisson-like firing [11]. Nonetheless, the common feature to all those input structures is the collective spiking behavior. Second, neuronal activity in cortex shows a considerable amount of variability even if the very same experimental paradigm is repeated multiple times [12], even though protocols with reliable responses were also observed [13]. Rate-based representations and learning rules address the issue of noisy inputs by averaging activity over time or across neurons, considering this variability as noise. Previous studies have proposed that this noise may have a functional role related to probabilistic representations of the environment in a Bayesian framework [14, 15]. However, the spiking variability has also been closely linked to behavior [16–18]. Experimental and theoretical evidence thus points to a relation between the variability of neuronal activity and the representation of the

stimulus. This is the basis for the present study, which aims to make a step toward an equivalent of STDP for supervised learning; for simplicity we study the new concept with non-spiking neurons.

These observations raise several questions: How can a neuronal system perform its function not despite this large amount of variability, but using variability itself? Consequently, how to train networks that employ representations based on variability such as covariances? Finally, one may wonder if covariance-based learning is superior to technical solutions that employ a mean-based representation, providing a reason why it may have evolved in neuronal circuits. To address these questions, we consider the training of a neuronal system that has to learn time series with structured variability in their activity.

Supervised learning in (artificial) neuronal networks is often formulated as a gradient descent for an objective function that measures the mismatch between the desired and the actual outputs [19]. The most prominent examples of such synaptic update rules are the delta rule for the "classical" perceptron that is a neuronal network with an input layer and an output layer [20–22] and error back-propagation for the multilayer perceptron [23]. These led to the modern forms of deep learning and convolutional networks [24, 25]. Their success was only unleashed rather recently by the increased computational power of modern computers and large amounts of available training data, both required for successful training. A key for further improvement of neuronal information processing lies on evolving the theory, for example by devising new and efficient paradigms for training.

A central feature of the training design is how the physical stimulus is represented in terms of neuronal activity. To see this, consider the classical perceptron whose task is to robustly classify patterns of input activities despite their variability within each category. For the case of two categories (or classes), it seeks a plane within the vector space of input activities that best separates the classes and the classification performance depends on the overlap between the two clouds of sample data points. Applied to time series, this paradigm can be used relying on the mean activity as the relevant feature of the input signals; the variances of the input signals that measure departures from the respective means are then akin to noise that might negatively affect the classification. For time-dependent signals, this scheme has been extended by considering as representative pattern for each category the mean trajectory over time (instead of the average activity as before); the variability then corresponds to meaningless fluctuations around the mean trajectory. This view has led to efficient technical solutions to train neuronal networks by recurrent back-propagation or by back-propagation through time [26–29].

We here present a novel paradigm that employs the covariances of cofluctuating activity to represent stimulus information, at the intersection between neuroscience and machine learning. We show how the input-output mapping for covariances can be learned in a recurrent network architecture by efficiently training the connectivity weights by a gradient-descent learning rule. To do so, we use an objective (or cost) function that captures the time-series variability via its second-order statistics, namely covariances. We derive the equivalent of the delta rule for this new paradigm and test its classification performance using synthetic data as well as moving digits in the visual field.

The remainder of the article is organized as follows: Section formalizes the concept behind our learning paradigm based on the stochastic fluctuations and contrasts it with distinct concepts studied previously like noise correlations. Section considers a network with feed-forward connectivity that is trained —following each stimulus presentation— to implement a desired mapping from the input covariance to the output covariance. To this end, we derive a gradient-descent learning rule that adapts the feed-forward connections and examine the network training in theory, for infinite observation time, as well as for time series of limited duration. Section extends the training of Section to a network with both afferent and recurrent

connections. We show how recurrent connections allow us to exploit the temporal structure of input covariances as an additional dimension for stimulus representation that can be mapped to output representations. Importantly, we demonstrate the specific role played by the recurrent network connectivity when the information to learn is in the temporal dimension of covariances, but not in its spatial dimension. Last, Section applies the covariance perceptron to moving digits, to illustrate its ability in capturing dynamic patterns in data closer to real-life signals.

## Time series and covariance patterns

Various types of correlations for time series have been studied in the literature, as illustrated in Fig 1A. We denote by $x_1^{t,s}$ and $x_2^{t,s}$ two time series, where the superscript $t$ indicates time and the superscript $s$ the trial index. The situation in the left column of Fig 1A corresponds to a stereotypical trajectory for $x_1^{t,s}$ across trials, which translates to positive correlation for two trials $s$ and $s'$:

$$\mathrm{corr}_t(x_1^{t,s}, x_1^{t,s'}) > 0, \tag{1}$$

and similarly for $x_2^{t,s}$. Here the subscript $t$ indicates the ensemble over which correlations are computed. We refer to this as 'signal correlation' because the "information" to learn is the stereotypical trajectory that can be evaluated by averaging over trials. Such reliable trajectories can be learned using back-propagation through time [29]. In contrast, the situation in the middle columns illustrates correlation within each trial between the two time series, either with zero lag for the same $t$

$$\mathrm{corr}_s(x_1^{t,s}, x_2^{t,s}) > 0, \tag{2}$$

or for distinct $t$ and $t'$ with a fixed lag $\tau = t' - t$

$$\mathrm{corr}_s(x_1^{t,s}, x_2^{t',s}) > 0. \tag{3}$$

Importantly, the middle plots illustrate that distinct trials may exhibit very different trajectories, even though the within-trial correlations are the same; the latter can thus be the basis of information conveyed by time series to be learned. This is the paradigm examined in the present study, which can be thought as cofluctuations of fast-varying firing rates that strongly interact with STDP [11]. It conceptually differs from another type of correlation that has been much studied, referred to as noise correlations [30, 31]. For time series, noise correlations concern the trial-to-trial correlation of the means of the time series, as represented by the horizontal dashed lines in the right plot of Fig 1A, formally given by

$$\mathrm{corr}_s(\langle x_1^{t,s} \rangle_t, \langle x_2^{t,s} \rangle_t) > 0, \tag{4}$$

where the angular brackets denote the average over time to compute the mean (it will be formally defined later).

In the context corresponding to the examples in the middle columns of Fig 1A, we consider the classification problem of discriminating time series. Their within-trial correlations, as defined in Eqs (2) and (3), which are the "information" to learn. It is worth noting that 'signal correlations' in the left column of Fig 1A can also lead to reliable correlation patterns in the sense of Eqs (2) and (3) depending on their average trajectories for pairs of inputs and on the time window used to calculate the correlations, as in the example that will be studied in Section. The general situation corresponds to the middle plot in Fig 1B, where two groups (in red

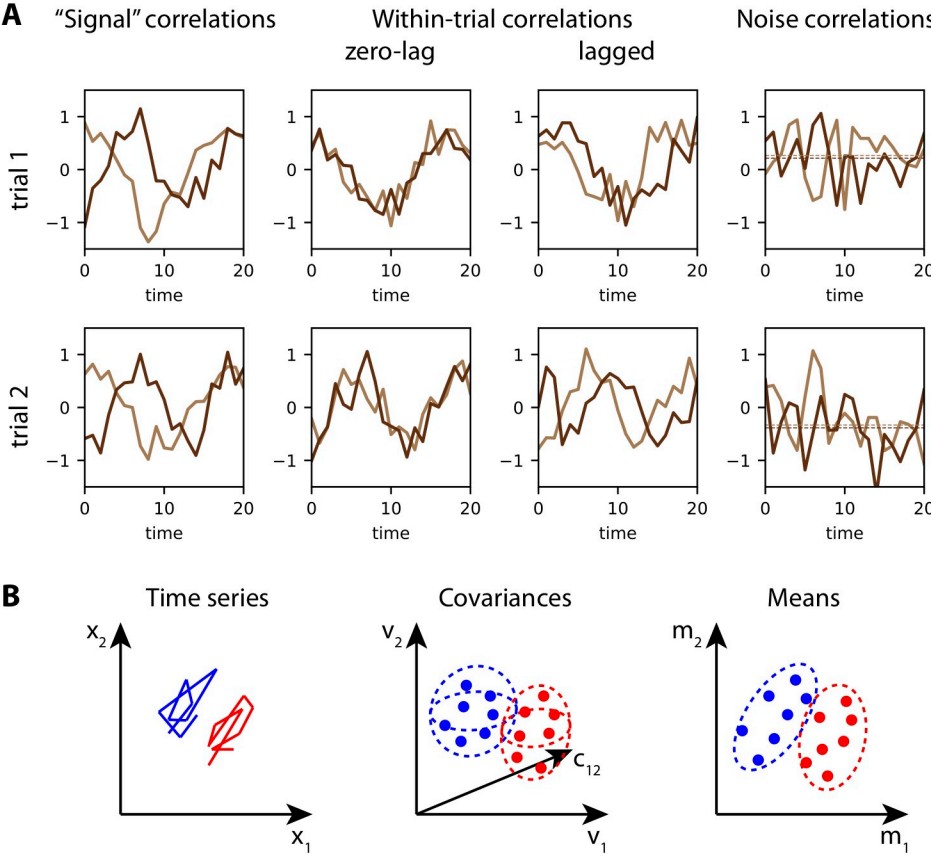

**Fig 1. Cofluctuations of time series as a basis for stimulus discrimination. A**: Three types of correlations for two time series (in light and dark brown). Signal correlations (left column) measure the similarity of individual trajectories across trials up to some additional noise. Importantly, the light and dark time series may be uncorrelated within each trial. Conversely, within-trial correlations (middle columns) correspond to the situation where trials may be distinct, but the two time series within a trial are correlated (positively here, the left configuration with zero lag and the right configuration with a lag of 3 for visual legibility). This is the subject of the present study. Last, noise correlations (right column) concern the means of the time series, as represented by the dashed lines, that are either both positive or both negative within each trial. **B**: For the discrimination of multivariate time series, as in panel A, we here consider two categories (red and blue). The time series $x_1^t$ and $x_2^t$, displayed in the $(x_1, x_2)$-plane in the left plot, show one example for each category. From each category example, one can calculate the mean (here a vector of dimension 2), corresponding to a single dot in the right plot. Learning for classification aims to find a separatrix between the red and the blue point clouds. The presence of noise correlations between the means affects the overlap between the dot groups (e.g. positive for the shown example), hence the classification performance. Alternatively, one can compute from the same time series their (within-trial) variances and covariances, yielding points in a three-dimensional space here (middle plot, where $v_1$ and $v_2$ are the respective variances, and $c_{12}$ the cross-covariance). Here classification is based on the within-trial covariances as features to learn, which conceptually differs from the mean-based learning and noise correlations in the right panel.

and blue, an example time series of each group being represented in the left plot) have distinct (co)variances that can be used as features for classification. In comparison, the right plot in Fig 1B depicts the equivalent situation where the means of the time series are used for discrimination. In this case noise correlations measure the spread of each dot cloud.

To implement the classification of time series based on their within-trial correlations, we examine the problem of the propagation of this signal in a neuronal network, as illustrated in Fig 2A. To deal with representations of stimulus identity embedded in temporal (co)fluctuations within trials, we move from the first-order statistics, the mean activity of each neuron within a trial, to the second-order statistics, the covariance between the fluctuating activities

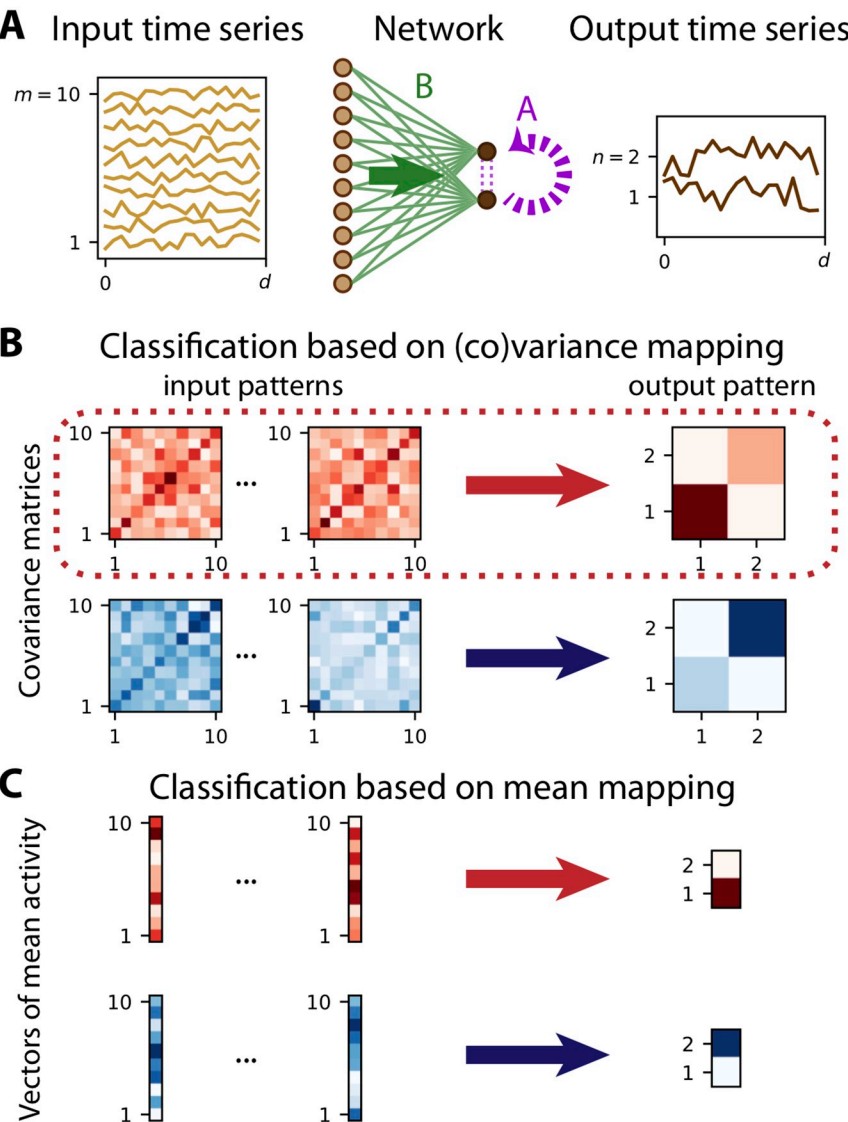

**Fig 2. From mean-based to covariance-based time-series classification. A**: Network with $n = 2$ output nodes generates a time series (in dark brown on the right) from the noisy time series of $m = 10$ input nodes (in light brown on the left). The afferent (feed-forward) connections $B$ (green links and green arrow) and, when existing, recurrent connections $A$ (purple dashed links and arrow) determine the input-output mapping. We observe the time series over a window of duration $d$. **B**: Each set of time series in panel A corresponds to a covariance pattern, namely an $m \times m$ matrix for the inputs on the left-hand side and an $n \times n$ matrix for the output on the right-hand side, where darker pixels indicate higher values. See Eq (7) for the formal definition of the averaging over the observation window of length $d$ in panel A. As an example, we define two categories (or classes) that are represented by larger variance of either of the two nodes, node 1 for the red category and node 2 for the blue category. The classification scheme is implemented by tuning the connectivity weights $A$ and $B$ such that several input covariance patterns are mapped to the single output covariance pattern of the corresponding category. **C**: As a comparison, considering the mean activities instead of the within-trial covariances, corresponds to the mapping between input and output vectors in Eq (6), which can be formalized in the context of the classical perceptron (linear or non-linear). There, the categories for the input pattern ($m$-dimensional vectors on the left-hand side) are defined by the output pattern ($n$-dimensional vector on the right-hand side), the red category with neuron 1 highly active and the blue category with neuron 2 highly active.

for pairs of neurons. To fix ideas, we consider a discrete-time network dynamics as defined by a multivariate autoregressive (MAR) process [32]. This linearization of neuron dynamics is to explore principles. The activity of the $m$ inputs $x_{1 \leq k \leq m}^t$ is described by a stochastic process in discrete time $t \in \mathcal{Z}$. The inputs drive the activity $y_{1 \leq i \leq n}^t$ of the $n$ output neurons via connections $B \in \mathbb{R}^{n \times m}$, which form the afferent connectivity. The outputs also depend on their own immediate past activity (i.e. with a unit time shift) through the connections $A \in \mathbb{R}^{n \times n}$, the recurrent connectivity, as

$$y_i^t = \sum_{1 \leq j \leq n} A_{ij} y_j^{t-1} + \sum_{1 \leq k \leq m} B_{ik} x_k^t \ , \tag{5}$$

illustrated in Fig 2A. We define the mean activities

$$\begin{aligned} X_k &\equiv \langle x_k^t \rangle_t \\ Y_i &\equiv \langle y_i^t \rangle_t \ , \end{aligned} \tag{6}$$

where the angular brackets $\langle \cdots \rangle_t = d^{-1} \sum_{t=1}^d \cdots$ indicate the average over the period of duration $d$ in Fig 2A. Likewise, the input and output covariances, with $\tau \in \mathcal{Z}$ being the time lag, are defined as

$$\begin{aligned} P_{kl}^\tau &\equiv \langle x_k^{t+\tau} x_l^t \rangle_t - \langle x_k^{t+\tau} \rangle_t \langle x_l^t \rangle_t \\ Q_{ij}^\tau &\equiv \langle y_i^{t+\tau} y_j^t \rangle_t - \langle y_i^{t+\tau} \rangle_t \langle y_j^t \rangle_t \ . \end{aligned} \tag{7}$$

Here we implicitly assume stationarity of the statistics over the observation window.

As a first step, we consider the case of vanishing means for covariance-based classification, so the second terms on the right-hand sides disappear in Eq (7); considerations about a mixed scenario based on both means and covariances will be discussed at the end of the article. In this setting, the goal of learning is to shape the mapping from the input covariance $P$ to the output covariance $Q$ in the network in Fig 2A in order to perform the task, here classification. The most general case would consider the mapping of the entire probability distributions. For the ensemble of Gaussian processes used here and the linear dynamics in Eq (5), the first two moments, however, uniquely determine the entire statistics. In the classification example, correlated fluctuations across neurons —as defined by covariances in Eq (7)— convey information that can be used to train the network weights and then classify input time series into categories. The desired outcome is illustrated in Fig 2B, where the 'red category' of input covariance matrices $P$ is mapped by the network to an output, where neuron 1 has larger variance than neuron 2. Conversely, for the 'blue category' of input covariances matrices, the variance of neuron 2 exceeds that of neuron 1. This example thus maps a bipartite set of patterns to either of the two stereotypical output patterns, each representing one class. In doing so, we focus on the mapping between input and output, on which a threshold is then applied to make the decision. This corresponds to the conditional probabilities relating input and output covariance patterns given by

$$\Pr\left(P^\tau | Q_{ij}^\tau > \theta\right) \propto \Pr\left(Q_{ij}^\tau > \theta | P^\tau\right) \Pr\left(P^\tau\right) \ . \tag{8}$$

We thus need to derive a learning scheme that tunes the connectivity weights, $A$ and $B$, to shape the output covariance when a given input covariance pattern is presented, since the input-output mapping governs $\Pr\left(Q_{ij}^\tau > \theta | P^\tau\right)$. We term this paradigm the 'covariance perceptron', since it can be seen as an extension of the classical perceptron in Fig 2C. In the covariance perceptron, the objective or cost function is changed to manipulate the covariance

of time series rather than the mean. Note that there is no non-linearity considered in the neuronal response here unlike what is typically used in the classical perceptron [21], which will be discussed later.

Importantly, our approach feeds the entire time series into the network, which outputs time series according to Eq (5). This embodies a mapping from the input covariances to the output covariances, which are defined in Eq (7) and evaluated in practice using an observation window. The discrimination of the time series based on their covariances thus results from the network dynamics itself. The parameters to tune are the $n^2 + nm$ synaptic weights $A$ and $B$. This is fundamentally different from a preprocessing of the data in the context of machine learning, where a set of features like covariances is extracted first and then fed to an (artificial) neuronal network that operates in this feature space (see Fig 1B). In the latter approach, each feature ($m(m + 1)/2$ for a zero-lag covariance matrix) would come with an individual weight to be tuned, then multiplied by the number $n$ of outputs. For classification where the input dimensionality $m$ is typically much larger that the number $n$ of categories, the use of resources (weights) is much lighter in our scheme. Another difference worth noting is that the measures on the input and output activities is of the same type in our scheme, so "information" is represented and processed in a consistent manner by the network. This opens the way to successive processing stages as in multilayer perceptrons.

Last, we stress again that our viewpoint on signal variability radically differs from that in Fig 2C, where the information is conveyed by the mean signal and fluctuations are noise. Conceptually, taking the second statistical order as the basis of information is an intermediate description between the detailed signal waveform and the (oversimple) mean signal. The switch from means to covariances implies that richer representations can be realized with the same number of nodes, thereby implementing a kernel trick [19] applied to time series using the network dynamics themselves.

## Learning input-output covariance mappings in feedforward networks

This section presents the concepts underlying the covariance perceptron with afferent connections $B$ only (meaning absent recurrent connectivity $A = 0$). For the classical perceptron in Fig 2C, the observed output mean vector $Y$ for the classification of the input mean vector $X$ defined in Eq (6) is given by the input-output mapping

$$X \mapsto Y = BX \ . \tag{9}$$

For time series, the derivation of this consistency equation —with $A = 0$ in Eq (5)— assumes stationary statistics for the input signals. Under the similar assumption of second-order stationarity, the novel proposed scheme in Fig 2B relies on the mapping between the input and output covariance matrices, $P^0$ and $Q^0$ in Eq (7), namely

$$P^0 \mapsto Q^0 = BP^0B^{\mathrm{T}} \ , \tag{10}$$

where T denotes the matrix transpose, and the superscript 0 denotes the zero time lag. Details can be found with the derivation of the consistency equation Eq (23) in Network dynamics (Methods). The common property of Eqs (9) and (10) is that both mappings are linear in the respective inputs ($X$ and $P^0$). However, the second is bilinear in the weight $B$ while the first is simply linear. Note also that this section ignores temporal correlations (i.e. we consider that $P^1 = P^{-1\mathrm{T}} = 0$); time-lagged covariances, in fact, do not play any role in Eq (23) when $A = 0$.

## Theory for tuning afferent connectivity based on spatial covariance structure

To theoretically examine covariance-based learning, we start with the abstraction of the MAR dynamics $P^0 \mapsto Q^0$ in Eq (10). As depicted in Fig 3A, each training step consists in presenting an input pattern $P^0$ to the network and the resulting output pattern $Q^0$ is compared to the objective $\bar{Q}^0$ in Fig 3B. For illustration, we use two categories (red and blue) of 5 input patterns each, as represented in Fig 3C and 3D. To properly test the learning procedure, noise is artificially added to the presented covariance pattern, namely an additional uniformly-distributed random variable with a magnitude of 30% compared to the range of the noiseless patterns $P^0$, independently for each matrix element while preserving the symmetry of zero-lag covariances; compare the noisy pattern in Fig 3A (left matrix) to its noiseless version in Fig 3C (top left matrix). The purpose is to mimic the variability of covariances estimated from a (simulated) time series of finite duration (see Fig 2), without taking into account the details of the sampling noise. The update $\Delta B_{ik}$ for each afferent weight $B_{ik}$ is obtained by minimizing the distance (see Eq (25) in Methods) between the actual and the desired output covariance

$$
\begin{aligned}
\Delta B_{ik} &= \eta_B \left( \bar{Q}^0 - Q^0 \right) \odot \frac{\partial Q^0}{\partial B_{ik}} \\
&= \eta_B \left( \bar{Q}^0 - Q^0 \right) \odot \left( U^{ik} P^0 B^{\mathrm{T}} + B P^0 U^{ik\mathrm{T}} \right) ,
\end{aligned}
\tag{11}
$$

where $U^{ik}$ is an $m \times m$ matrix with 0s everywhere except for element $(i, k)$ that is equal to 1; this update rule is obtained from the chain rule in Eq (26), combining Eqs (27) and (30) with $P^{-1} = 0$ and $A = 0$ (see Theory for learning rules in Methods). Here $\eta_B$ denotes the learning rate and the symbol $\odot$ indicates the element-wise multiplication of matrices followed by the summation of the resulting elements —or alternatively the scalar product of the vectorized matrices. Note that, although this operation is linear, the update for each matrix entry involves $U^{ik}$ that selects a single non-zero row for $U^{ik} P^0 B^{\mathrm{T}}$ and a single non-zero column for $B P^0 U^{ik\mathrm{T}}$. Therefore, the whole-matrix expression corresponding to Eq (11) is different from $(\bar{Q}^0 - Q^0) P^0 B^{\mathrm{T}} + B P^0 (\bar{Q}^0 - Q^0)^{\mathrm{T}}$, as could be naively thought.

Before training, the output covariances are rather homogeneous as in the examples of Fig 3C and 3D (initial $Q^0$) because the weights are initialized with similar random values. During training, the afferent weights $B_{ik}$ in Fig 3E become specialized and tend to stabilize at the end of the optimization. Accordingly, Fig 3F shows the decrease of the error $E^0$ between $Q^0$ and $\bar{Q}^0$ defined in Eq (25). After training, the output covariances (final $Q^0$ in Fig 3C and 3D) follow the desired objective patterns with differentiated variances, as well as small cross-covariances.

As a consequence, the network responds to the red input patterns with higher variance in the first output node, and to the blue inputs with higher variance in the second output (top plot in Fig 4B). We use the difference between the output variances in order to make a binary classification. The classification accuracy corresponds to the percentage of output variances with the desired ordering. The evolution of the accuracy during the optimization is shown in Fig 4C. Initially around chance level at 50%, the accuracy increases on average due to the gradual shaping of the output by the gradient descent. The jagged evolution is due to the noise artificially added to the input covariance patterns (see the left matrix in Fig 3A), but it eventually stabilizes around 90%. The network can also be trained by changing the objective matrices to obtain positive cross-covariances for red inputs, but not for blue inputs (Fig 4D); in that case variances are identical for the two categories. The output cross-covariances have separated distributions for the two input categories after training (bottom plot in Fig 4E), yielding the good classification accuracy in Fig 4F.

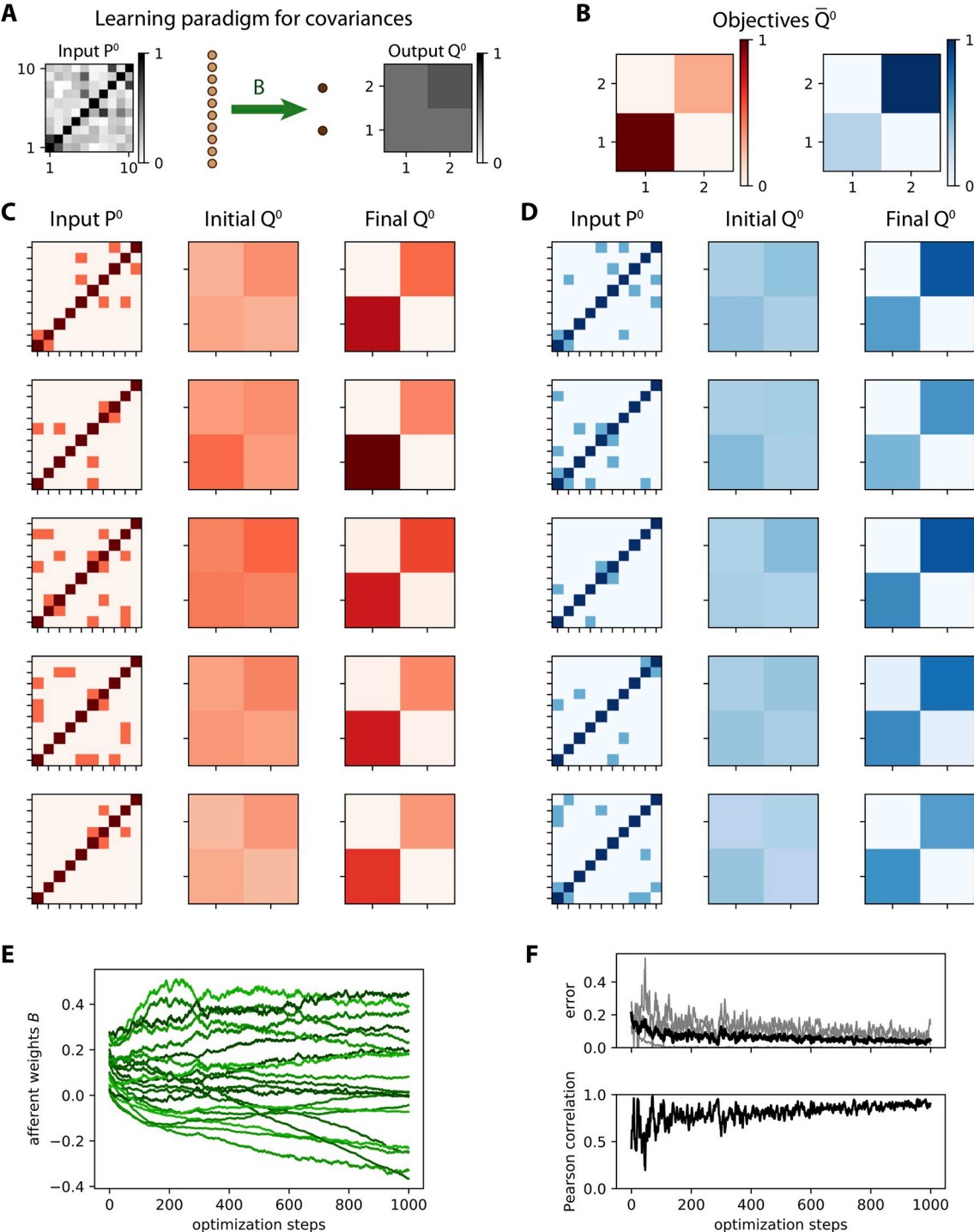

**Fig 3. Learning variances in a feed-forward network. A**: Schematic representation of the input-output mapping for covariances defined by the afferent weight matrix $B$, linking $m = 10$ input nodes to $n = 2$ output nodes. **B**: Objective output covariance matrices $\bar{Q}^0$ for two categories of inputs. **C**: Matrix for the 5 input covariance patterns $P^0$ (left column) for the first category, with their images under the original connectivity (middle column) and the final images after learning (right column). The training leads to a larger variance (darker pixel) for output 1 (bottom left pixel of matrices in the right column) than for output 2 (top right pixel). **D**: Same as C for the second category. The training leads to a larger variance for output 2 than for output 1 except for the last pattern. **E**: Evolution of individual weights of matrix $B$ during ongoing learning. **F**: The top panel displays the evolution of the error between $Q^0$ and $\bar{Q}^0$ at each step. The total error taken as the matrix distance $E^0$ in Eq (25) is displayed as a thick black curve, while individual matrix entries are represented by gray traces. In the bottom panel the Pearson correlation coefficient between the vectorized $Q^0$ and $\bar{Q}^0$ describes how they are "aligned", 1 corresponding to a perfect linear match.

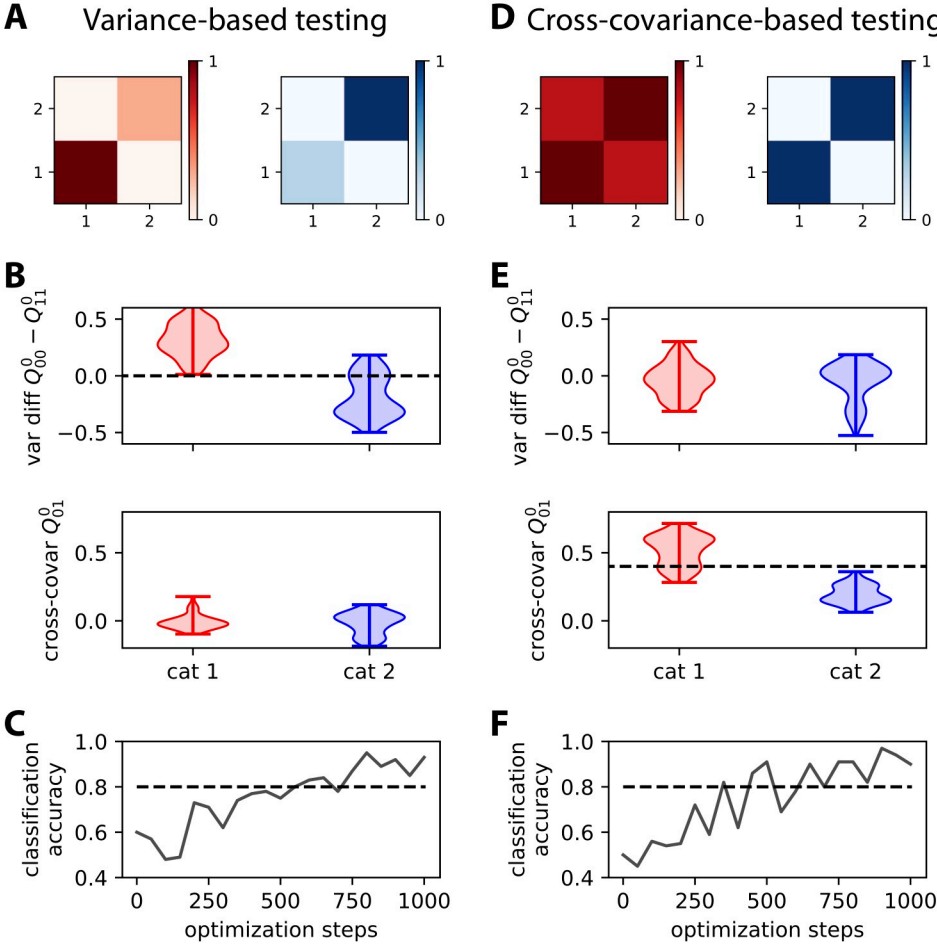

**Fig 4. Comparison between learning output patterns for variance and cross-covariance. A**: The top matrices represent the two objective covariance patterns of Fig 3B, which differ by the variances for the two nodes. **B**: The plots display two measures based on the output covariance: the difference between the variances of the two nodes (top) and the cross-covariance (bottom). Each violin plot shows the distributions for the output covariance in response to 100 noisy versions of the 5 input patterns in the corresponding category. Artificial noise applied to the input covariances (see the main text about Fig 3 for details) contributes to the spread. The separability between the red and blue distributions of the variances indicates a good classification. The dashed line is the tentative implicit boundary enforced by learning using Eq (30) with the objective patterns in panel A: Its value is the average of the differences between the variances of the two categories. **C**: Evolution of the classification accuracy based on the difference of variances between the output nodes during the optimization. Here the binary classifier uses the difference in output variances, predicting red if the variance of the output node 1 is larger than 2, and blue otherwise. The accuracy eventually stabilizes above the dashed line that indicates 80% accuracy. **D-F**: Same as panels A-C for two objective covariance patterns that differ by the cross-covariance level, strong for red and zero for blue. The classification in panel F results from the implicit boundary enforced by learning for the cross-covariances (dashed line in panel E), here equal to 0.4 that is the midpoint between the target cross-covariance values (0.8 for read and 0 for blue).

As a sanity check, the variance does not show a significant difference when training for cross-covariances (top plot in Fig 4E). Conversely, the output cross-covariances are similar and very low for the variance training (bottom plot in Fig 4B). These results demonstrate that the afferent connections can be efficiently trained to learn categories based on input (co)variances, just as with input vectors of mean activity in the classical perceptron.

## Discriminating time series observed using a finite time window

Now we turn back to the configuration in Fig 2A and verify that the learning procedure based on the theoretical consistency equations also works for simulated time series. This means that the sampled activity of the network dynamics itself is presented, rather than their statistics embodied in the matrices $P^0$ and $Q^0$, as done in Figs 3 and 4 and as a classical machine-learning scheme would do with a preprocessing step that converts time series using kernels. Again, the weight update is applied for each presentation of a pattern such that the output variance discriminates the two categories of input patterns. The setting is shown in Fig 5A, where only three input patterns per category are displayed.

To generate the input time series, we use a superposition of independent Gaussian random variables $z_l^t$ with unit variance (akin to white noise), which are mixed by a coupling matrix $W$:

$$x_k^t = \sum_{1 \leq l \leq m} W_{kl} z_l^t \ . \tag{12}$$

We randomly draw 10 distinct matrices $W$ with a density of $f = 10\%$ of non-zero entries, so the input time series differ by their spatial covariance structure $P^0 = WW^T$. At each presentation, one of the 10 matrices $W$ is chosen to generate the input time series using Eq (12). Their covariances are then computed using an observation window of duration $d$. The window duration $d$ affects how the empirical covariances differ from their respective theoretical counterpart $P^0$, as shown in Fig 5C. This raises the issue of the precision of the empirical estimates required in practice for effective learning.

As expected, a longer observation duration $d$ helps to stabilize the learning, which can be seen in the evolution of the error in Fig 5D: the darker curves for $d = 20$ and 30 have fewer upside jumps than the lighter curve for $d = 10$. To assess the quality of the training, we repeat the simulations for 20 network and input configurations ($W$ and $z$, resp.), then calculate the difference in variance between the two output nodes as in Fig 4B and 4C. Training for windows with $d \geq 20$ achieve very good classification accuracy in Fig 5E. This indicates that the covariance estimate can be evaluated with sufficient precision from only a few tens of time points. Moreover, the performance only slightly decreases for denser input patterns (Fig 5F). Similar results can be obtained while training the cross-covariance instead of the variances.

## Discrimination capacity of covariance perceptron for time series

The efficiency of the binary classification in Fig 4 relies on tuning the weights to obtain a linear separation between the input covariance patterns. Now we consider the capacity of the covariance perceptron, evaluated by the number $p$ of input patterns (or pattern load) that can be discriminated in a binary classification. For the classical perceptron and for randomly-chosen binary patterns that must be separated in two categories, the capacity is $2m$, twice the number $m$ of inputs [33, 34]. An analytical study of the capacity of the covariance perceptron using Gardner's theory of connections from statistical mechanics is performed in a sister article [35]. That study shows that a single readout cross-covariance can in theory discriminate twice as many patterns per synapse as the classical perceptron, but that this capacity does not linearly scale with the number of outputs. Finding such optimal solutions is an NP-hard problem in general and optimization methods like the gradient descent employed here may only achieve suboptimal capacities in practice. In addition, an important difference compared to that study, which focused on "static" patterns, concerns the time series used for training and testing here, which involves empirical noise (see Fig 5C). Thus, we here employ numerical simulation to get a first insight on the capacity of the covariance perceptron with "noisy inputs", varying in the same manner the number of input neurons and patterns to learn.

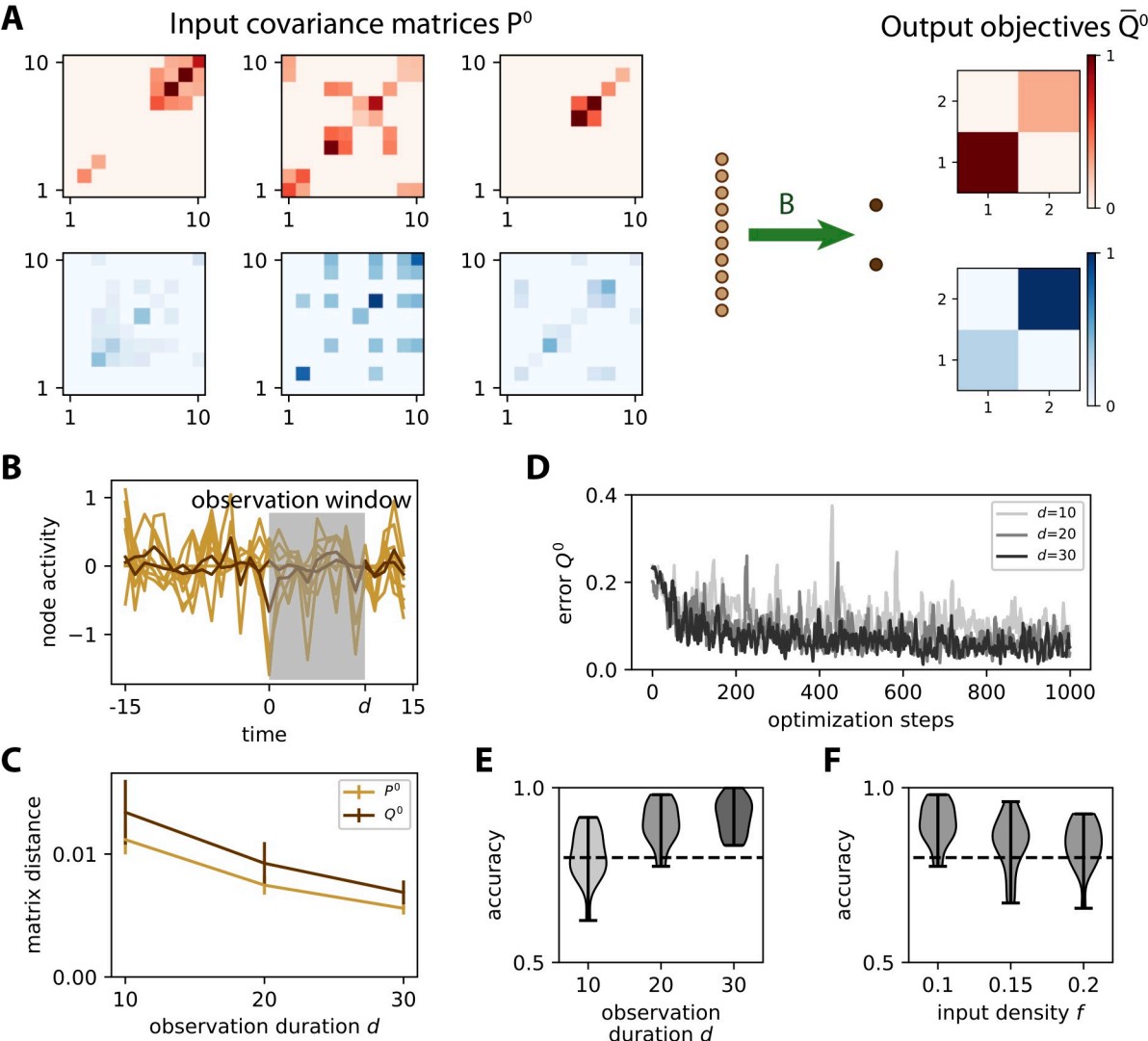

**Fig 5. Learning input covariances by tuning afferent connectivity. A**: The same network as in Fig 3A is trained to learn the input spatial covariance structure $P^0$ of time series governed by the dynamics in Eq (12). Only 3 matrices $P^0 = WW^T$ out of the 5 for each category are displayed. Each entry in each matrix $W$ has a probability $f = 10\%$ of being non-zero, so the actual $f$ is heterogeneous across the different matrices $W$. The objective matrices (right) correspond to a specific variance pattern for the output nodes. **B**: Example of simulation of the time series for the inputs (light brown traces) and outputs (dark brown). An observation window (gray area) is used to calculate the covariances from simulated time series. **C**: Sampling error as measured by the matrix distance between the covariance estimated from the time series (see panel B) and the corresponding theoretical value when varying the duration $d$ of the observation window. The error bars indicate the standard error of the mean over 100 repetitions of randomly drawn $W$ and afferent connectivity $B$. **D**: Evolution of the error for 3 example optimizations with various observation durations $d$ as indicated in the legend. **E**: Classification accuracy at the end of training (cf. Fig 4C) as a function of $d$, pooled for 20 network and input configurations. For $d \geq 20$, the accuracy is close to 90% on average, mostly above the dashed line indicating 80%. **F**: Similar plot to panel E when varying the input density of $W$ from $f = 10$ to 20%, with $d = 20$.

Here we consider the same optimization of output variances on which the discrimination is based as in Fig 5, instead of the cross-covariances studied using Gardner's theory [35]. The evolution of the classification accuracy averaged over 10 configurations is displayed in Fig 6A, where darker gray levels correspond to larger network sizes as indicated in the legend. For each configuration, the mean accuracy of the last three epochs is plotted in Fig 6B where the observation duration is $d = 20$. At the load $p = m$, the performance decreases with the network size: for instance, it remains in the case of $m = 100$ inputs (darker curve) around 75% for a

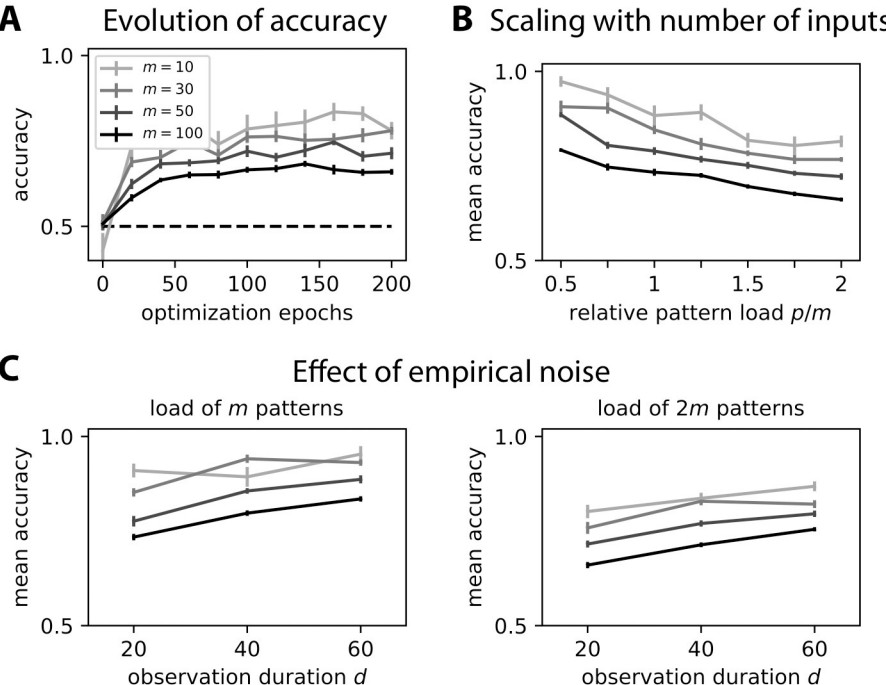

**Fig 6. Numerical evaluation of capacity. A**: Evolution of the classification accuracy over the optimization epochs. During each epoch, all $p = m$ patterns are presented in a random order. We use the same network as in Fig 5, but with distinct input numbers $m$ as indicated in the legend. The observation duration is $d = 20$. The error bars correspond to the standard error of the mean accuracy over 10 configurations. **B**: Comparison of the classification accuracies as a function of the relative pattern load $p/m$ (x-axis). Note that 2 corresponds to the theoretical capacity $p = 2m$ of the classical perceptron with a single output, but the architecture considered here has 2 outputs; for an in-depth study of the capacity and its scaling with the number of output nodes, please refer to our sister paper [35] whose results are discussed in the main text. The plotted values are the mean accuracies for each configuration, averaged over the last three epochs in panel A. The error bars indicate the standard error of the mean accuracy over 10 repetitions for each configuration. **C**: Similar plot to panel B when varying the observation duration $d$ for two cases $p = m$ and $p = 2m$.

load of $m$ patterns and way above 50% for $2m$ patterns. Interestingly, the performance significantly increases when using larger $d$, for example improving by roughly 10% for $m = 50$ and 100 in each of the two plots of Fig 6C. This means that the empirical noise related to the covariance estimation over the observation window (see Fig 5B) becomes larger when the number $m$ of inputs increases, but it can nonetheless be compensated by using a larger window.

## Learning spatio-temporal covariance mapping with both afferent and recurrent connectivities

We now extend the learning scheme of Section to the tuning of both afferent and recurrent connectivities in Eq (5) with the same application to classification. We also switch from spatial to spatio-temporal covariance structures, corresponding to a non-zero lag $\tau$ in Eq (7). As a model of input, we simulate time series that differ by their hidden dynamics. By "hidden dynamics" we simply mean that time series obey a dynamical equation, which determines their spatio-temporal structure that can be used for classification. Concretely, we use

$$x_k^t = \sum_l W_{kl} x_l^{t-1} + z_k^t \;, \tag{13}$$

with $z_k^t$ being independent Gaussian random variables of unit variance. This dynamical

equation replaces the superposition of Gaussians in Eq (12) for generating temporally correlated input signals, where $P^0$ satisfies the discrete Lyapunov equation $P^0 = WP^0W^T + 1_m$. Here, $1_m$ is the identity matrix, and $P^1 = WP^0$ denotes the 1-lag covariances. In this context, a category consists of a set of such processes, each with a given matrix $W$ in Eq (13) as before with $P^0$ in Fig 3. Note that the matrix $W$ itself is not known to the classifier, only the resulting statistics of $x$ that obeys Eq (13); thus we call this setting "classification of hidden dynamics".

## Stability of ongoing learning

Before examining classification, we consider the problem of stability of ongoing learning (or plasticity). Unsupervised Hebbian learning applied to recurrent connections is notoriously unstable and adequate means to prevent ongoing plasticity from leading to activity explosion are still under debate [36, 37] —note that, if those studies concern especially spiking networks, their conclusions also apply to non-spiking networks as considered here. Supervised learning, however, can lead to stable weight dynamics [38, 39]. Stability can be directly enforced in the objective function, but can also be a consequence of the interplay between the learning and network dynamics. Because our objective functions are based on the output covariances, we test whether they also yield stability for the weights and network activity.

The learning procedure is tested with simulated time series as in Fig 5. The weight updates are given by equivalent equations to Eq (11) that determine the weight updates for the afferent and recurrent connectivities, $B$ and $A$ respectively; see Eqs (30), (32), (33) and (34) in Theory for learning rules in Methods. We recall that they rely on the consistency equations (23) and (24), which are obtained in Network dynamics (Methods) under the assumption of stationary statistics. Fig 7 illustrates the stability of the learning procedure, while the error decreases to the best possible minimum. As a first example, we want to map 10 input patterns —corresponding to 10 distinct matrices $W$ in Eq (13), each giving a specific pair of input covariance matrices $(P^0, P^1)$— to the same objective covariance matrix $\bar{Q}^0$, thereby dealing with a single category as illustrated in Fig 7A. Note that with the choice of $W$ with small weights here, all input covariance matrices $P^0$ are close to the identity and the optimized connectivity must generate cross-correlations between the outputs. Adapting Eqs (26) and (27) in Methods to the current configuration, the weight updates are given by

$$
\begin{aligned}
\Delta A_{ij} &= \eta_A \left(\bar{Q}^0 - Q^0\right) \odot \frac{\partial Q^0}{\partial A_{ij}} \ , \\
\Delta B_{ik} &= \eta_B \left(\bar{Q}^0 - Q^0\right) \odot \frac{\partial Q^0}{\partial B_{ik}} \ ,
\end{aligned}
\tag{14}
$$

where the derivatives are given by the matrix versions of Eqs (30) and (32) in Theory for learning rules (Methods):

$$
\begin{aligned}
\frac{\partial Q^0}{\partial A_{ij}} &= A\frac{\partial Q^0}{\partial A_{ij}}A^T + V^{ij}Q^0A^T + AQ^0V^{ijT} + V^{ij}BP^{-1}B^T + BP^{-1T}B^TV^{ijT} \ , \\
\frac{\partial Q^0}{\partial B_{ik}} &= A\frac{\partial Q^0}{\partial B_{ik}}A^T + U^{ik}P^0B^T + BP^0U^{ikT} + AU^{ik}P^{-1}B^T + ABP^{-1}U^{ikT} \\
&\quad + U^{ik}P^{-1T}B^TA^T + BP^{-1T}U^{ikT}A^T \ .
\end{aligned}
\tag{15}
$$

Both formulas have the form of a discrete Lyapunov equation that can be solved at each optimization step to evaluate the weight updates for $A$ and $B$. Also recall that the derivation of the consistency equations in Network dynamics (Methods) assumes $P^2 = 0$ and is thus an

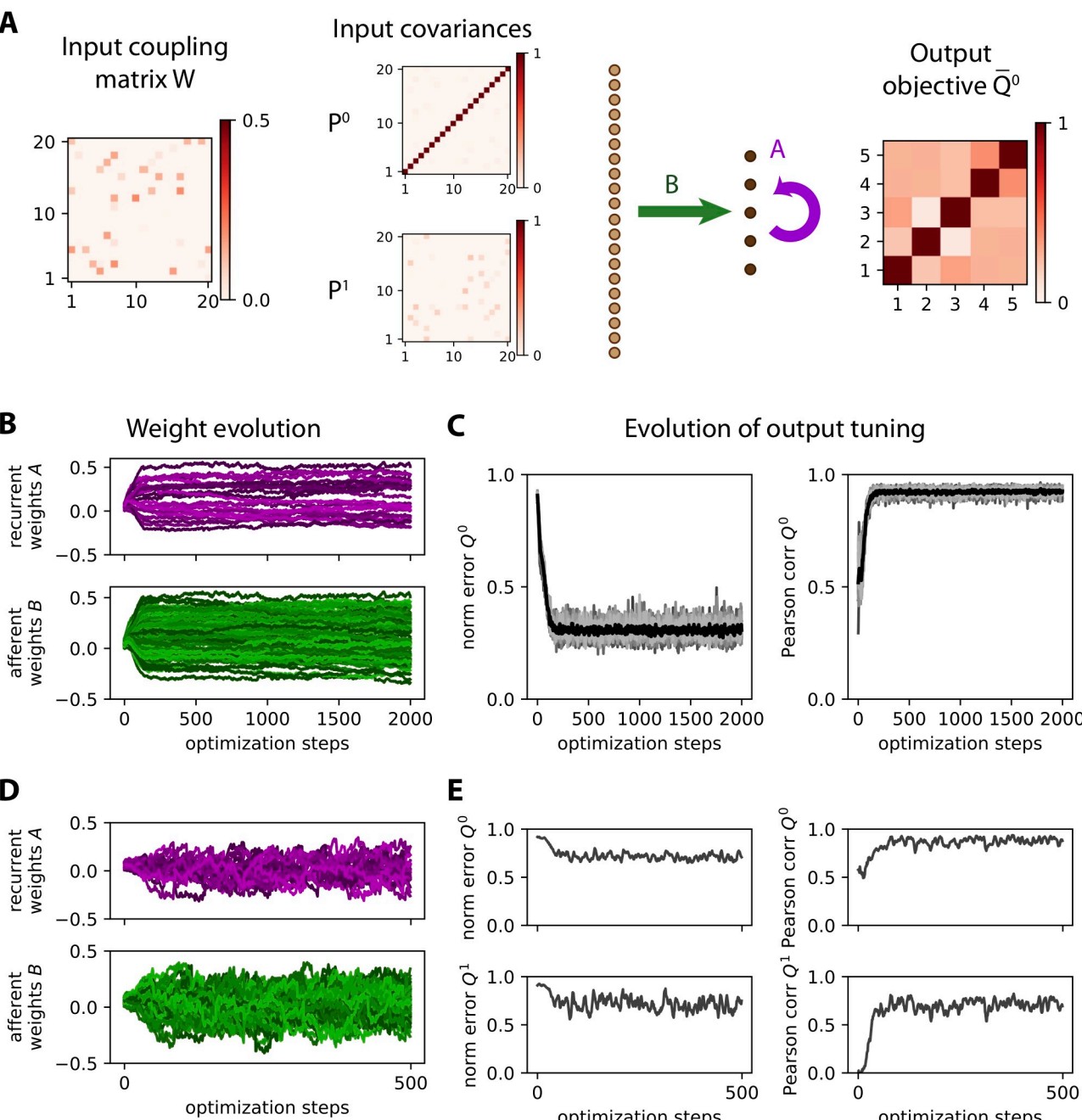

**Fig 7. Stability of ongoing learning for both afferent and recurrent connectivities. A**: Network configuration with $m = 20$ inputs and $n = 5$ outputs. Each input pattern corresponds to a randomly chosen $m \times m$ matrix $W$ with 10% density of non-zero connections that determines the input covariance matrices $(P^0, P^1)$ of the time series generated from Eq 13. The objective $\bar{Q}^0$ is a randomly-drawn $m \times m$ symmetric matrix (also ensuring its definite positivity). Here we consider a single category (in red) to test whether the weight learning rule can achieve a desired input-output mapping, leaving aside classification for a moment. **B**: Example evolution of the afferent and recurrent weights (green and purple traces, respectively). Simulation of the time series as in Fig 5, observed for a duration $d = 50$. The network has to map 10 input pattern pairs $(P^0, P^1)$ similar to the example in panel A to a single output $\bar{Q}^0$. **C**: Evolution of the tuning of the network output. The learning procedure aims to reduce the normalized error between the output covariance $Q^0$ and its objective $\bar{Q}^0$ (left plot, similar to Fig 3F), here calculated as $\| \bar{Q}^0 - Q^0 \| / \| \bar{Q}^0 \|$ where $\|\cdots\|$ is the matrix norm. We also compute the Pearson correlation coefficient between the vectorized matrices $Q^0$ and $\bar{Q}^0$ (right plot). The thick black traces correspond to the mean over 10 repetitions of similar optimizations to panel A, each gray curve corresponding to a repetition. **D-E**: Similar plots to panels A-B for the tuning of both $(\bar{Q}^0, \bar{Q}^1)$ for a single repetition. The input time series are generated in the same manner as before. Contrary to panel A, the output objective pair $(\bar{Q}^0, \bar{Q}^1)$ is chosen as two homogeneous diagonal matrices with larger values for $\bar{Q}^0$ than $\bar{Q}^1$, corresponding to outputs with autocorrelation and no cross-correlation —this example is inspired by previous work on whitening input signals [40]. The observation duration is $d = 100$.

approximation because we have $P^2 = W^2 P^0$ here. As the input matrix $W$ must have eigenvalues smaller than 1 in modulus to ensure stable dynamics, our approximation corresponds to $\|P^2\| = \|WP^1\| \ll \|P^1\|$. The purpose of this example is thus to test the robustness of the proposed learning in that respect. The weight traces appear to evolve smoothly in Fig 7B. In the left plot of Fig 7C, the corresponding error between the output $Q^0$ and the objective $\bar{Q}^0$ firstly decreases and then stabilizes. The evolution of the Pearson correlation (right plot of Fig 7C) further indicates that the matrix structure of the output $Q^0$ remains very close to that of $\bar{Q}^0$, once it reached saturation, even though the network may not perfectly converge towards the objectives as indicated by the residual error.

A second and more difficult example is explored in Fig 7D and 7E, where the objective is a pair of matrices $\bar{Q}^0$ and $\bar{Q}^1$. The weight optimization then involves the equivalent of Eq (15) for $Q^1$. The issue of whether there exists a solution for the weights $A$ and $B$ to implement the desired mapping is more problematic because the defined objectives imply many constraints, namely Eqs (23) and (24) must be satisfied for all ten pairs $(P^0, P^1)$ with $(Q^0, Q^1) = (\bar{Q}^0, \bar{Q}^1)$ with the same weight matrices. This results in less smooth traces for the weights (Fig 7D) and a weak decrease for the normalized error (left plot in Fig 7E), suggesting that there is not even an approximate solution for the weights $A$ and $B$. Nonetheless, the weight structure does not explode and the Pearson correlation between the output covariances and their respective objectives indicate that the objective structure is captured to some extent (right plot in Fig 7E).

## Computational and graph-local approximations of the covariance-based learning

First, we consider an approximation in the calculation of the weight updates that does not require solving the Lyapunov equation. An important question is which role the elements that quantify the non-linearity due to the recurrent connectivity $A$ play in determining the weight updates. As explained around Eq (36) in Methods, we consider the approximation of Eq (15) that ignores second-order terms in the recurrent connectivity matrix $A$ in the Lyapunov equation

$$
\begin{aligned}
\frac{\partial Q^0}{\partial A_{ij}} &= V^{ij}Q^0A^{\mathrm{T}} + AQ^0V^{ij\mathrm{T}} + V^{ij}BP^{-1}B^{\mathrm{T}} + BP^{-1\mathrm{T}}B^{\mathrm{T}}V^{ij\mathrm{T}} \;, \\
\frac{\partial Q^0}{\partial B_{ik}} &= U^{ik}P^0B^{\mathrm{T}} + BP^0U^{ik\mathrm{T}} + AU^{ik}P^{-1}B^{\mathrm{T}} + ABP^{-1}U^{ik\mathrm{T}} \\
&\quad + U^{ik}P^{-1\mathrm{T}}B^{\mathrm{T}}A^{\mathrm{T}} + BP^{-1\mathrm{T}}U^{ik\mathrm{T}}A^{\mathrm{T}} \;.
\end{aligned}
\tag{16}
$$

Now the calculation of the weight updates is much simpler, involving only a few matrix multiplications and additions. To test the validity of this approximation, we repeat the same optimization as in Fig 7A with 10 input patterns to map to a single output pattern with objective $\bar{Q}^0$. As illustrated for an example in Fig 8A, the comparison of Eq (16) (red trace) with Eq (15) (black trace) hardly show any difference in the performance. This is confirmed in Fig 8B for 10 repetitions of the same training with randomly chosen input and output patterns. Although the approximation for the solution of the Lyapunov equation may seem coarse, it yields a very similar trajectory for the gradient descent. This gives the intuition that this computational approximation gives the correct general direction in the high-dimensional space and, since the weight updates are computed at each optimization step and these steps are small, the gradient descent does not deviate from the "correct" direction. Even though this example involves non-full connectivity where only about 30% of the weights are trained (others being kept equal to zero), the same simulation with full connectivity (not shown) gives

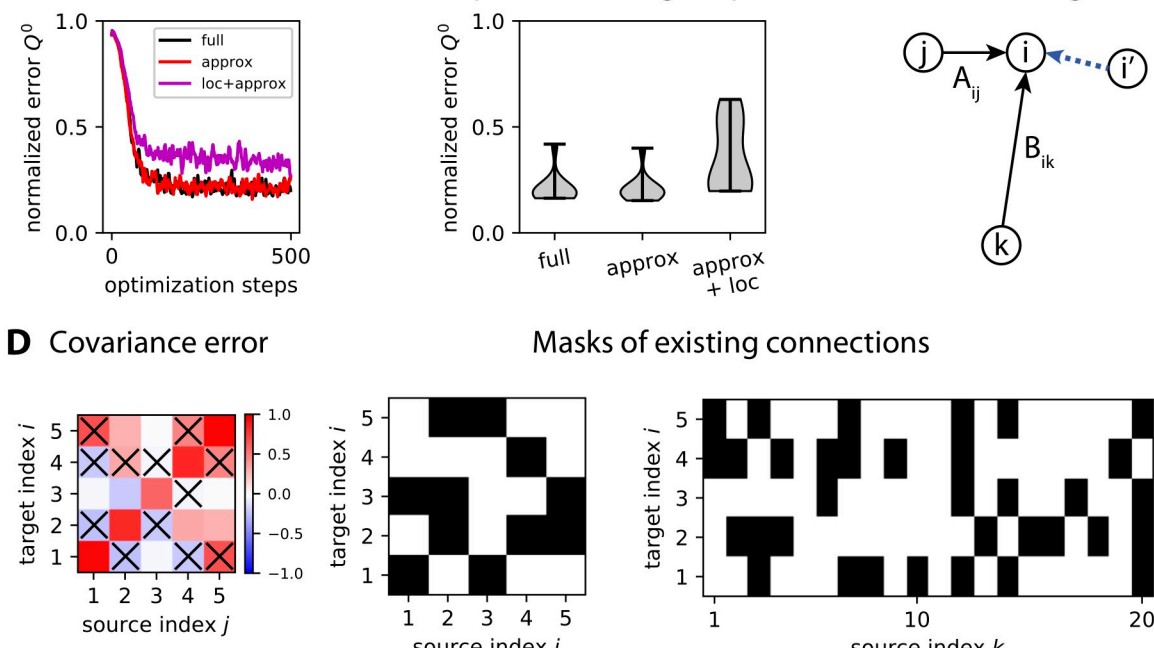

**Fig 8. Approximations of the gradient descent. A**: Comparison of the evolution of the error in optimizing $Q^0$ for three flavors of the gradient descent: the "full" solution (in black) using Eq (15), the computational approximation (in red) using Eq (16) and the local approximation (in purple) using Eq (17). Network and pattern configuration similar to Fig 7A, involving 10 input patterns to map to a single output pattern by training both afferent and recurrent connections in a network of $m = 20$ inputs and $n = 5$ outputs. Here the network has sparse connectivity, corresponding to a probability of existence for each connection equal to 30%; weights for absent connections are not trained and kept equal to 0 at all times. **B**: Asymptotic error estimated from the last 10 optimization steps in panel A for 10 repetitions of similar configurations to panel A. **C**: Schematic representation of the local approximation in Eq (17) to compute the weight updates of afferent connections and recurrent connections targeting neuron $i$ (here with two examples $B_{ik}$ and $A_{ij}$): only covariances from network neurons with a connection to neuron $i$ (like the "parent" neuron $i'$ via the dashed blue arrow) are taken into account. **D**: Example of network connectivity (binary matrices in black on the right) that determine which elements of the covariance error matrix (in color on the left) are used to calculate the weight update in Eq (17). Crosses in the left matrix indicate discarded elements, that correspond to absent recurrent connections. Note that variances are never discarded.

similar results with no distinguishable difference between the full computation and the computational approximation.

Second, we consider a local approximation where the information necessary to compute the weight updates is only accessible from presynaptic neighbor neurons in the network as illustrated in Fig 8C:

$$
\begin{aligned}
\Delta A_{ij} &= \eta_A \sum_{i' \in S_i} \left( \bar{Q}^0_{ii'} - Q^0_{ii'} \right) \frac{\partial Q^0_{ii'}}{\partial A_{ij}} \;, \\
\Delta B_{ik} &= \eta_B \sum_{i' \in S_i} \left( \bar{Q}^0_{ii'} - Q^0_{ii'} \right) \frac{\partial Q^0_{ii'}}{\partial B_{ik}} \;,
\end{aligned}
\tag{17}
$$

where $S_i$ is the subset of "parent" neurons with a connection to neuron $i$. The rationale here is that information related to the activities of a neuron pair can be evaluated at the point of contact that are recurrent synapses. Of course, this only makes a difference in the case of non-full connectivity (around 30% density in Fig 8) and this local approximation requires the computational approximation since solving the Lyapunov equation in the full calculation of Eq (15) requires the knowledge of all connections within the network. Using the expressions in

Eq (16) for recurrent connections while ignoring afferent connections, we have

$$
\begin{aligned}
\Delta A_{ij} &= \eta_A \sum_{i' \in S_i} (\bar{Q}^0_{ii'} - Q^0_{ii'}) \left( 2 \sum_{j'} Q^0_{jj'} A_{i'j'} \right) , \\
&= \eta_A \sum_{i' \in S_i} (\bar{Q}^0_{ii'} - Q^0_{ii'}) \, 2 \left\langle y^t_j \left( \sum_{j' \in S_{i'}} A_{i'j'} y^t_{j'} \right) \right\rangle ,
\end{aligned}
\tag{18}
$$

This means that the necessary information for computing the weight update is the summed activity received by each parents neuron $i' \in S_i$, which has to be centralized by the downstream $i$. The same observation is valid for the summed activity received by each neuron $i'$ from the input neurons. This approximation is local in the graph in the sense that it only uses information from neighbor (parent) neurons. An example of the matrix elements in the covariance error that contribute to the weight update is displayed in Fig 8D. Although the performance decreases compared to the computational approximation as illustrated in Fig 8A and 8B, this local optimization still performs reasonably well.

## Classification of time series with hidden dynamics

From the dynamics described in Eq (5), a natural use for $A$ is the transformation of input spatial covariances ($P^0 \neq 0$ and $P^1 = 0$) to output spatio-temporal covariances ($Q^0 \neq 0$ and $Q^1 \neq 0$), or vice-versa ($P^0 \neq 0$, $P^1 \neq 0$, $Q^0 \neq 0$ and $Q^1 = 0$). S1 Appendix provides examples for these two cases that demonstrate the ability to tune the recurrent connectivity together with the afferent connectivity (from now on with 100% density), which we further examine now.

We consider input time series that are temporally correlated and spatially decorrelated, meaning that $P^1$ conveys information about the input category (i.e. reflecting the hidden dynamics), but not $P^0$. The theory predicts that recurrent connectivity is necessary to extract the relevant information to separate the input patterns. To our knowledge this is the first study that tunes recurrent connectivity in a supervised manner to specifically extract temporal information from lagged covariances when spatial covariances are not informative about the input categories. Concretely, we here use 6 matrices $W$ (3 for each category) to generate the input time series that the network has to classify based on the output variances, illustrated in Fig 9A. Importantly, we choose $W = \exp(\mu \mathbf{1}_m + V)$ with exp being the matrix exponential, $V$ an antisymmetric matrix and $\mu < 0$ for stability. As a result, the zero-lag covariance of the input signals $P^0 = \frac{1}{1 - e^{2\mu}} \mathbf{1}_m$ is exactly the same for all patterns of either category, proportional to the identity matrix as illustrated in Fig 9B. This can be seen using the discrete Lyapunov equation $P^0 = W P^0 W^T + \mathbf{1}_m$, which is satisfied because $WW^T = \exp(2\mu \mathbf{1}_m + V + V^T) = e^{2\mu} \mathbf{1}_m$. In contrast, the time-lagged covariances $P^1 = W P^0$ differ across patterns, which is the basis for distinguishing the two categories.

The output is trained only using $Q^0$ according to Eq (15), meaning that the input spatio-temporal structure is mapped to an output spatial structure —also following the above considerations about the existence of adequate weights to implement the desired input-output covariance mapping. The covariances from the time series are computed using an observation window of duration $d$ in the same manner as before in Fig 5B. Note that it is important to discard an initial transient period to remove the influence of initial conditions on both $x^t$ and $y^t$. In practice, we use a larger window duration $d$ compared to Fig 5, as it turns out that the output covariances are much noisier here. The influence of $d$ can also be seen in Fig 9C, where the evolution of the error for the darkest curves with $d \geq 60$ remain lower on average than the lighter curve with $d = 20$. To assess the quality of the training, we repeat the simulations for 20

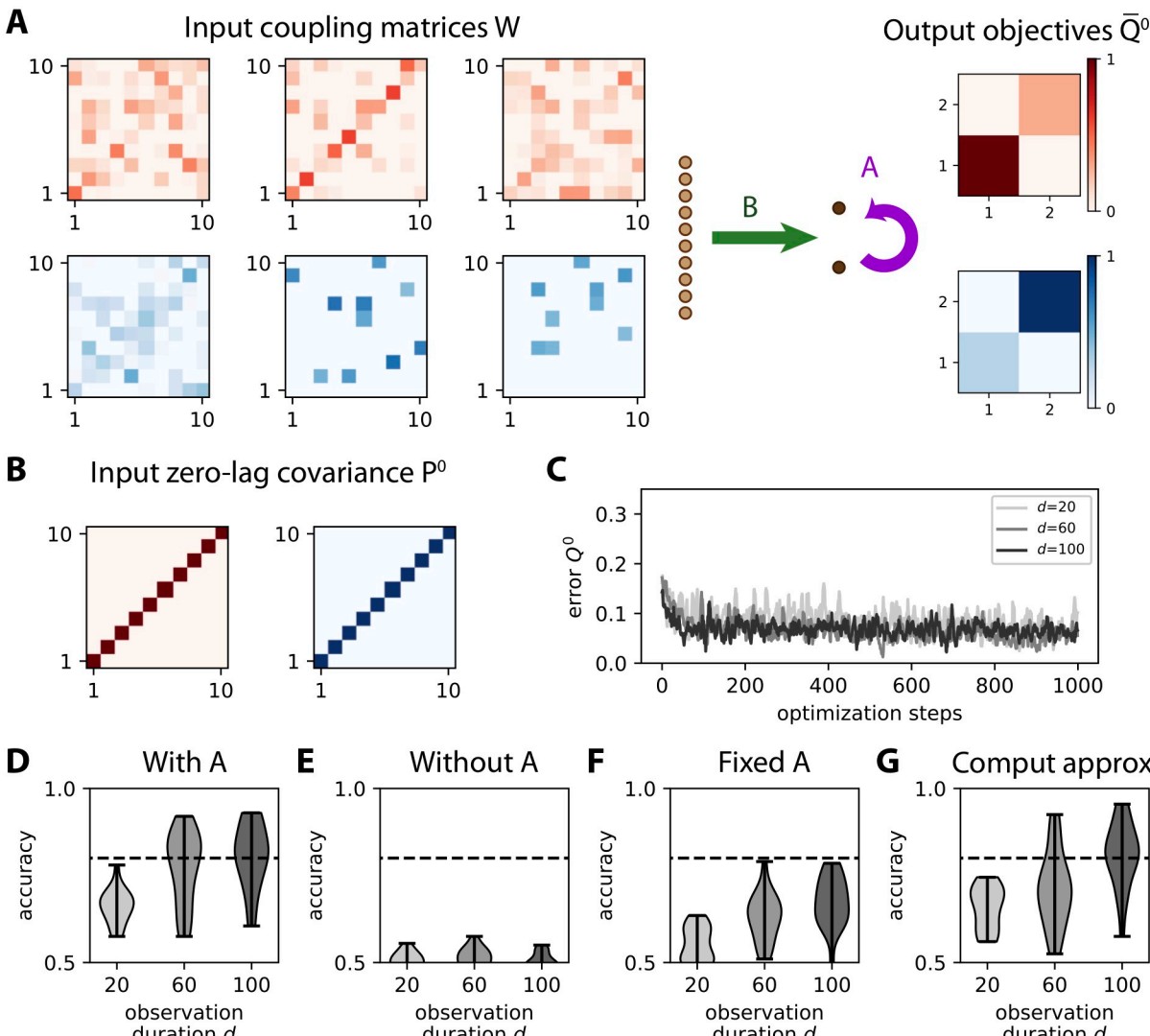

**Fig 9. Learning input spatio-temporal covariances with both afferent and recurrent connectivities. A**: Network architecture with $m = 10$ input nodes and $n = 3$ output nodes, the latter being connected together by the recurrent weights $A$ (purple arrow). The network learns the input spatio-temporal covariance structure, which is determined here by a coupling matrix $W$ between the inputs as in Eq (13). Here we have 3 input patterns per category. The objective matrices (right) correspond to a specific variance for the output nodes. **B**: The matrices $W$ are constructed such that they all obey the constraint $P^0 \propto \mathbf{1}_m$. **C**: Evolution of the error for 3 example optimizations with various observation durations $d$ as indicated in the legend. **D**: Classification accuracy after training averaged over 20 network and input configurations. For the largest $d = 100$, the accuracy is above 80% on average (dashed line). The color contrast corresponds to the three values for $d$ as in panel C. **E**: Accuracy similar to panel D with no recurrent connectivity ($A = 0$). **F**: Same as panel D with a random fixed matrix $A$ and switching off its learning. **G**: Same as panel D with the computational approximation in Eq (16) that does not require solving the Lyapunov equation.

network and input configurations and then calculate the difference in variance between the two output nodes for the red and blue input patterns. The accuracy gradually improves from $d = 20$ to 100 in Fig 9D. When enforcing $A = 0$ in Fig 9E, classification stays at chance level. This is expected and confirms our theory, because the learning for $B$ only captures differences in $P^0$, which is the same for all patterns here. When $A$ is sufficiently large (but fixed), it contributes to some extent to $Q^0$ such that the weight update for $B$ can extract relevant information as can be seen in Eq (15), raising the performance above chance level in Fig 9F. Nonetheless, the performance remains much worse than when the recurrent connections are optimized. These

results demonstrate the importance of training recurrent connections in transforming input spatio-temporal covariances into output spatial covariances. Last, Fig 9G shows that the computational approximation in Eq (16) performs as well as the full gradient in Eq (15) that involves solving the Lyapunov equation for this task.

## Application to the recognition of moving digits

Finally, we examine the robustness of the covariance perceptron applied to the recognition of objects that move in the visual field by a network of sensory (input) and downstream (output) neurons. To this end, we use the MNIST database of handwritten digits 0 to 4 [41]. As illustrated with the digit 0 in Fig 10, each digit moves horizontally in the visual field either to the right or the left. The goal is to train readout neurons to detect both the digit identity and the motion direction. The digits pass through the receptor fields of two "vertical columns" of input neurons ($m = 18$), which results in delayed activity between the columns (light and dark brown in Fig 10B). For each digit, the traces "viewed" by an input neuron exhibit large variability across presentations, see Fig 10C for digits 0 and 2 moving right. The goal of this demonstration is not so much to find the best classification tool, but to see how our proposed learning and classification scheme performs with "real-life" data, in particular when the Gaussian assumption for inputs is violated. Note that we use in this section the non-centered moments instead of the centered moments that are the rigorous definition of the covariances. The image of the digit is swept over a two-dimensional receptor array. As a result, information about both motion direction and digit identity is transformed into only spatial covariances between the inputs. In the example of Fig 10D, we see distinct mean patterns for the left and right moving digit 0 (see the upper left and lower right quadrants), as well as yet another pattern for digit 2. In other words, both variances and cross-covariances are required for correct classification, the former being strongly related to the digit identity and the latter to the motion direction. We now test whether the covariance perceptron can efficiently extract the relevant information from the second-order statistics of those patterns, while comparing it to other classification networks.

The confusion matrices in Fig 10E represent the predictions for the training set throughout the optimization procedure using Eq (11). For each moving digit (a category for each row), the diagonal matrix element increases over epochs, corresponding to the improvement of the classification performance. Conversely, off-diagonal elements become smaller, indicating a reduction of the prediction errors. Importantly, the same is true for the test set in Fig 10F. The similarity between the confusion matrices underlines that the covariance perceptron generalizes very well to unseen samples of the same category, which is crucial for practical applications.

We vary the number of samples in the training and in the test set and repeat the procedure in Fig 10 to further evaluate the robustness of our covariance-based classification. The evolution of the classification performance is displayed in Fig 11A for 500 to 50000 training samples (light to dark gray curves). The classification performance improves faster with more samples, but appears to saturate for 5000 and 50000 training samples at the same value, around 71%. Importantly, the test accuracy is equal to the training accuracy when sufficiently many samples are used, indicating an absence of overfitting and a good generalization to unseen data.

A technical point here is discarding output cross-covariances in the training to only tune the output variances that are used for the classification. This improves the classification performance by more than 10% in the example of Fig 11B (gray versus green traces). The result can be intuitively understood by the fact that cross-covariances add further constraints on the weights. In particular, enforcing zero cross-covariances between the output means

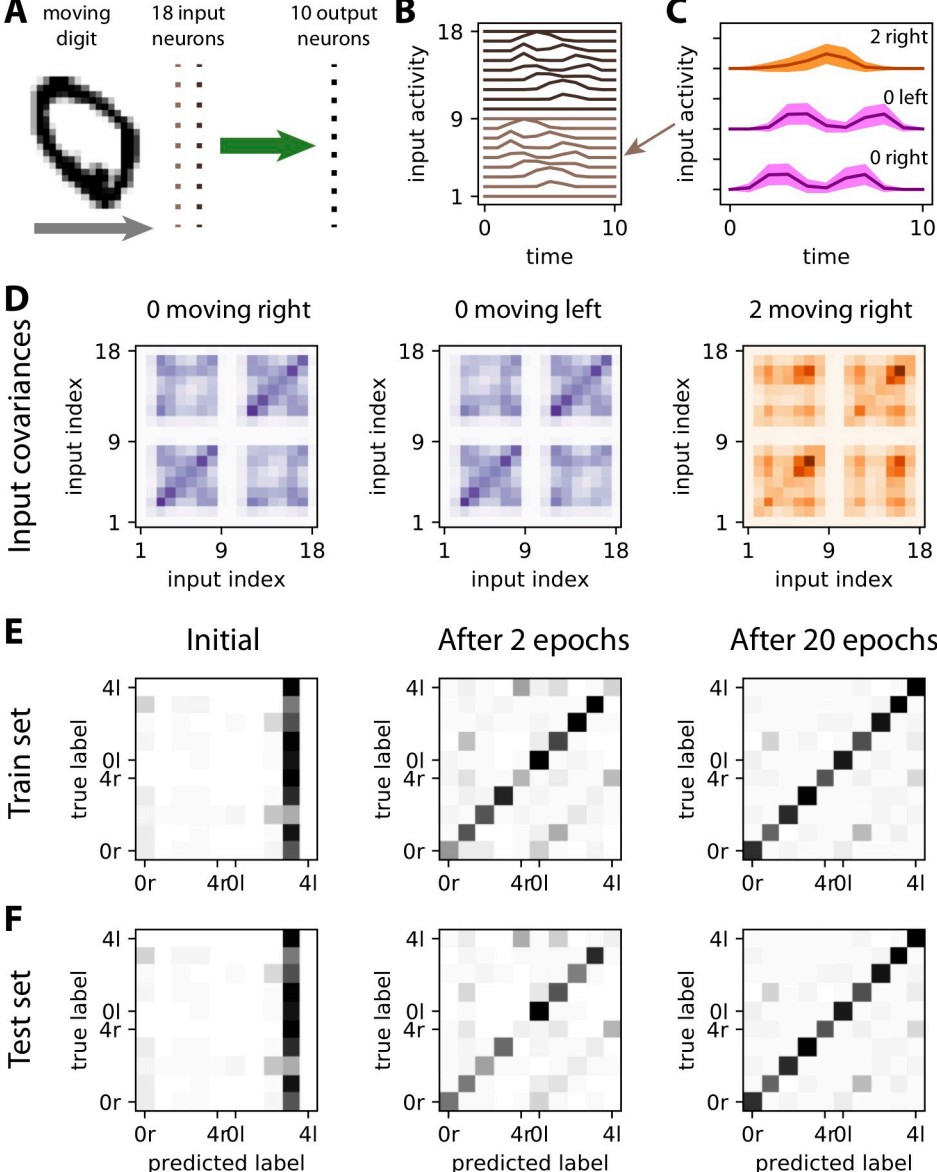

**Fig 10. Learning moving digits. A**: Moving digit in the visual field, where the input neurons are represented at the center of their respective receptor fields (vertical lines of dark and light brown dots). Each input neuron has a receptor field of $3 \times 3$ pixels, which amounts to downscaling the MNIST dataset from its original size $28 \times 28$ to $9 \times 9$. The 10 output neurons (one per category, the largest output variance indicates the predicted category) only have afferent connections (in green), which are trained using Eq (11) as in Fig 3. **B**: Responses of the input to the digit 0 in panel A moving to the right. The activity of the neurons in the left column (indexed from 1 to 9, in light brown) increases before that for the neurons of the right column (from 10 to 18, in dark brown). **C**: Mean activity traces for input neuron 5 (see arrow in panel B) for 10 samples of digits 0 and 2, both moving left or right as indicated above. The colored areas correspond to the standard deviation of the time-varying activity over all patterns of each category. **D**: The information relative to the moving stimulus is reflected by specific patterns in the input covariance matrices $P^0$ (averaged over all patterns of each category), left for moving from the left and right for moving from the right for digit 0. Differences are located in the cross-covariances between neurons from different columns (upper left and lower right quadrants). The covariance structure is also digit specific, as illustrated by the comparison with digit 2. **E**: Confusion matrices of the covariance perceptron for the train set (5000 samples with digits 0 to 4, balanced over the categories) before learning, after 2 epochs and after 20 epochs. The category labels indicate the digit and the direction ('r' for right and 'l' for left). As before, the classification is based on the higher variance among two outputs, one per category (digit and motion). Diagonal matrix elements correspond to correct classification, others to errors. **F**: Same as panel E for the test set (500 samples with different digits from the train set).

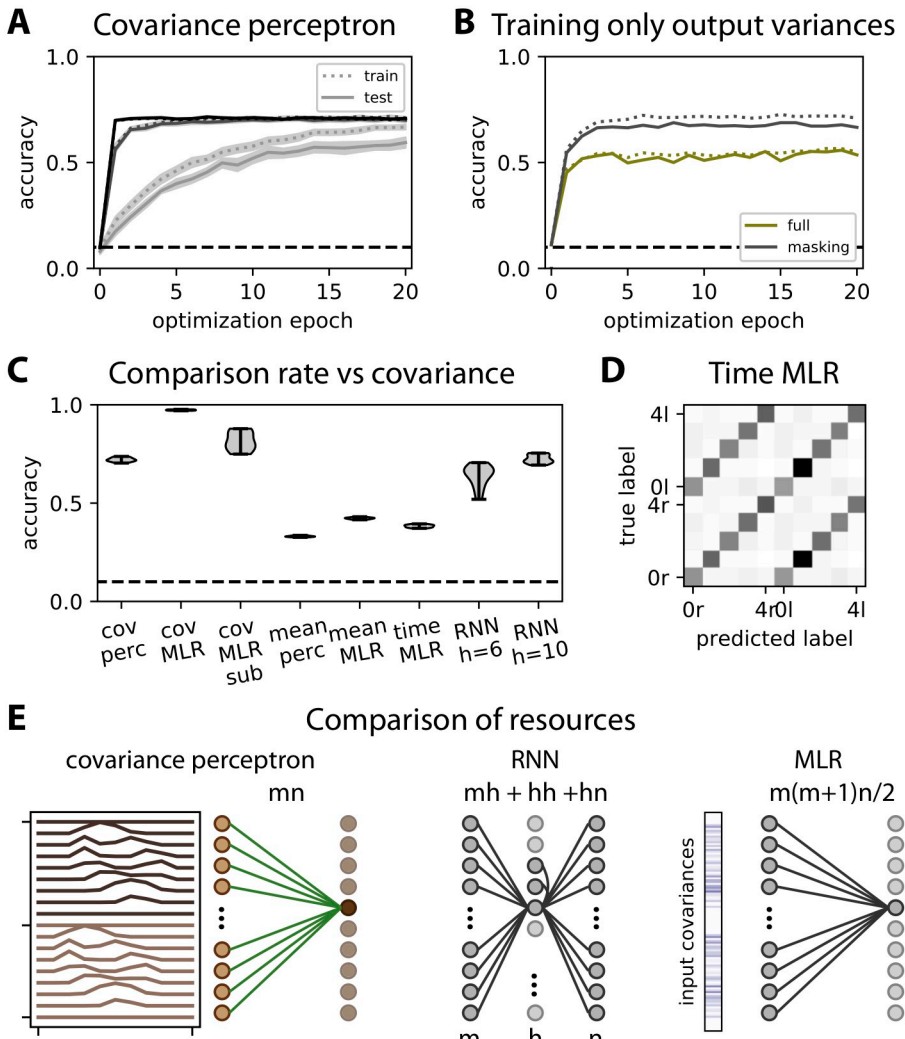

**Fig 11. Comparison with machine learning techniques like the classical perceptron.** As in Fig 10, five digits from 0 to 4 are considered with left or right motion, yielding 10 categories. **A**: Evolution of the performance of the covariance perceptron during training for various sizes for train and test sets: the light to dark gray curves correspond to 500, 5000, and 50000 patterns, respectively. The test set has 10 times fewer samples (50, 500 and 5000, resp.). During each epoch, all respective patterns are used for training and testing. The curves indicate the mean over 10 repetitions and the surrounding area the corresponding standard error of the mean (which is in fact small); the solid and dotted curves correspond to the train and test sets (see legend). The dashed horizontal line indicates chance-level accuracy. **B**: Comparison between the accuracy of the covariance perceptron when training the full output covariance matrix ('full') or only the variances on the diagonal ('masking') that are used for the classification of categories. The traces correspond to an example with 5000 training samples and the accuracies for the train and test sets are represented as in panel A. **C**: Comparison of classification accuracy between the covariance perceptron ('cov perc'), mean-based perceptron processing time series ('mean perc'), the multinomial logistic regression (MLR) that corresponds to the classical perceptron with a sigmoidal function to implement its non-linearity and the recurrent neural network (RNN). The mean perceptron and the covariance perceptron have the same architecture with 18 inputs and 10 outputs. The RNN has 18 inputs, 10 outputs and either $h = 6$ or 10 hidden neurons, the first version involving roughly as many weights as the covariance perceptron. The RNN-based classification relies on the mean activity of the output neurons calculated over an observation window (see main text for details). The MLR is trained in three configurations: using the mean patterns over the observation window for train and test ('mean MLR'); covariance patterns for train and test ('cov MLR'); time samples that are passed to the non-linear sigmoid before being averaged over the observation period in order to capture second-order statistics as described in the main text ('time MLR'). **D**: Confusion matrix for for the time MLR, corresponding to the test set at the end of learning. Note the errors on the secondary diagonal within the upper left and lower right quadrants, corresponding to error in classifying the direction. **E**: Schematic diagrams of the networks' configuration for the covariance perceptron (left), the RNN (middle) and the MLR applied to covariance patterns (right). The total numbers of weights (or optimized parameters) to tune for each classifier is indicated above the connections, in terms of number $m = 18$ of inputs in the time series and $n = 10$ outputs for each category, plus $h = 6$ or 10 hidden neurons for the RNN.

decorrelating them "spatially", which is not primarily useful for classification. Therefore, we use some masking here to only retain information about the output variances in $\bar{Q}^0 - Q^0$ when calculating the weight updates.

The performance for 50000 test samples is summarized in Fig 11C ('cov perc') and compared with a classical machine-learning tool, multinomial logistic regression (MLR) that corresponds to the classical non-linear perceptron with a sigmoidal function, applied to the same covariance input patterns ('cov MLR'). The MLR for patterns of covariances is quasi perfect at 98%, confirming the expected outcome that the covariances provide all the necessary information to classify the digits with their motion direction. Beside the non-linearity, a main reason for the difference in the performances between the covariance perceptron and the MLR is that the MLR uses many more resources: a regression coefficient (equivalent to a weight to optimize, we ignore the bias here) per element of the input covariance matrix $P^0$, as illustrated in Fig 11E, totaling $m(m + 1)n/2 = 1530$ weights to train since $m = 18$ and $n = 10$ here after taking into account the symmetry of $P^0$. In contrast, the covariance perceptron only uses $mn = 180$ afferent weights per output for the classification. To match the number of resources, we repeat the MLR classification by randomly subsampling 180 distinct matrix elements of $P^0$, reducing the performance to 81%. The performance of the covariance perceptron is thus about 10% lower than that of the MLR with matched resources, which presumably comes from the non-linearity of the MLR. To further check the importance of the non-linearity, we repeat the experiment with a linear regressor instead of the MLR and the performance becomes 84% using the full covariance matrix and 1530 weights to train, but drops to 48% for matched resources with 180 weights. This indicates that the covariance perceptron makes efficient use of resources for the classification based on a linear mapping between the input and output covariances.

We then compare the previous results to classification procedures based on mean output activity. The linear version of the classical perceptron corresponds to the network architecture in Fig 10A where the learning rule is based on patterns corresponding to mean activity of each input over the observation window, as described around Eq (39) in Methods, gives a performance of 33%. In contrast, the performance of the MLR, corresponding to the classical perceptron with a non-linear sigmoidal function, applied to the same patterns is 42% (supposedly thanks to the non-linearity). For both classifiers, the left and the right directions are not distinguished. This is expected because the two directions give the same mean for the input over the observation period, see the two examples for digit 0 in Fig 10C. However, the results may be different when considering the output activity of the network in Fig 10A at each time point of the observation window with the classical perceptron with a non-linearity determined by a function $\phi$. Considering the general situation of a MLR with given coefficients $w$ (equivalent to the weights $B$), the response at each time point of the presented stimulus $x^t$ is given by $y^t = \phi(w^T x^t)$. If the data $x^t \in \mathbb{R}^N$ are distributed with some distribution $p(x^t)$, the output of the network also depends on potentially all moments of this distribution, which can be seen in the Taylor expansion of the non-linearity. This leads to the mean output over the observation window $\langle y^t \rangle = \int p(x^t)\,\phi(w^T x^t)\,dx^t \simeq \phi(0) + \phi'(0)w^T\langle x^t\rangle + \frac{1}{2}\phi''(0)\,w^T\langle x^t x^{tT}\rangle\,w + \ldots$ for a Taylor expansion around the value 0, where the angular brackets correspond to the general averaging over the distribution of $x^t$. Then, the last term is related to the covariance matrix $P^0 = \langle x^t x^{tT}\rangle_t - \langle x^t\rangle_t\langle x^t\rangle_t^T$ where the angular brackets are defined as in Eqs (6) and (7); the relevant information here corresponds to the mean trajectories in Fig 10C and the noise is to be understood as the variability over the samples of the same categories (note that this can be related to the 'signal correlations' in the left column of Fig 1A). The difference between this application of the MLR and our paradigm is, though, that the mean and the covariance

additively contribute to the mean of the output activity. Our setup allows us to consider these two terms in isolation. Regarding the mapping by the network activity pointwise in time, non-linearities can thus also be regarded as a mechanism that transforms information contained in the covariance into information in the mean. Applied to the present case of time series, the information needed for the classification is in the covariances $\langle x_k x_l^{\mathrm{T}} \rangle_t \simeq \int x_k^t x_l^t \mathrm{d}t$ where the input product is integrated over the observation window. We can thus train a MLR to perform the classification based on the mean output activity, averaged over the observation period $\langle y_i \rangle_t \simeq \int y_i^t \mathrm{d}t$. If the input covariances can be significantly captured thanks to the non-linearity, then the MLR should be able to discriminate between the two directions. We find, however, a similar performance for the MLR trained using Eq (42) with category-specific objectives that correspond to constant activity over the window (see 'time MRL' in Fig 11C), where the motion direction is not well captured as indicated by the the confusion matrix in Fig 11D. This suggests that the first-order statistics override the second-order statistics in the learning procedure. This confirms that the qualitative difference of our approach from a direct application of classical machine-learning tools also has practical implications in terms of classification performance.

Last, we compare the same classification with a recurrent neural network (RNN) trained by basic back-propagation through time (BPTT) to test in a different manner whether the mean trajectories as those in Fig 10C can be used for prediction. The learning rules apply with $L = 5$ steps backward in time and the network comprises additional output neurons from the recurrently connected hidden neurons, as described in Fig 11E. The objective of the training is for the RNN to have a larger activity in the output neuron corresponding to the stimulus category. Details are given in Methods, see Eqs (44) to (47). Among the two implementations of the RNN, the version with $h = 6$ involves roughly the same number of weights to optimize than the covariance perceptron, namely $mh + hh + hn = 204$ as illustrated in Fig 11E, and yields a poorer performance of 64% compared to the covariance perceptron. The other version with $h = 10$ corresponding to $mh + hh + hn = 380$ weights yields 73%, only slightly better than that of the covariance perceptron. Further comparison with refinements of RNN like long-short-term memory units is left for future work, in particular to explore with these moving digits which measure applied to the input time series yield the best discrimination.

## Discussion

This paper presents a new learning theory for the tuning of input-output mappings of a dynamic linear network model by training its afferent and recurrent connections in a supervised manner. The proposed method extracts regularities in the spatio-temporal fluctuations of input time series, as quantified by their covariances. As an application, we showed how this can be used to categorize time series: networks can be trained to map several input time series to a stereotypical output time series that represents the respective category, thus implementing a 'covariance perceptron'. We stress that, beyond the application to classification, our results can be regarded as information compression for the input patterns and our theory could also be used for other supervised learning schemes like autoencoders.

The conceptual change of perspective compared to many previous studies is that variability in the time series is here the basis for the information to be learned, namely the second-order statistics of the co-fluctuating inputs. This view, which is based on dynamical features, thus contrasts with classical and more "static" approaches that consider the variability as noise, potentially compromising the information conveyed in the mean activity of the time series. Beyond asking whether time series can be robustly classified despite their variability, we instead provide a positive answer to the question if variability can even be employed to

represent information in its covariance structure. Importantly, covariance patterns can involve time lags and are a genuine metric for time series, describing the propagation of activity between nodes. In contrast to the application of a measure to time series as a preprocessing step for machine-learning algorithms like the perceptron, our scheme opens the door to a self-consistent formulation of information processing of time series in recurrent networks, where the source signal and the classifier output have the same structure.

A main result is that the covariance perceptron can be trained to robustly classify time series with various covariance patterns, while observing a few time points only (Fig 5). For practical applications, the transformation of dynamical information about stimuli into spatial covariances that can be learned turns out to be powerful, as illustrated for the detection of both digit identity and motion direction with the MNIST digits (Figs 10 and 11). Importantly, our covariance-based detection can be robustly implemented by networks with limited resources (number of weights to train, see Fig 11). The other main result is the demonstration that the covariance perceptron can classify time series with respect to their hidden dynamics, based on temporal covariance structure only (Fig 9). Taken together, these results demonstrate that the covariance perceptron can distinguish the statistical dependencies in signals that obey different dynamical equations.

## Covariance-based decoding and representations

The perceptron is a central concept for classification based on artificial neuronal networks, from logistic regression [19] to deep learning [24, 25]. The mechanism underlying classification is the linear separability of the input covariance patterns performed by a threshold on the output activity, in the same manner as in the classical perceptron for vectors. All "dimensions" of the output covariance can be used as objectives for the training, cross-covariances and variances in $Q^0$, as well as time-shifted covariances in matrix $Q^1$. As with the classical perceptron, classification relies on shaping the input-output mapping, for example by potentiating afferent weights from an input with high variance to two outputs to generate correlated activity between the outputs. Note that, in general, the existence of an achievable mapping between the input patterns and the objective patterns is not guaranteed, even when tuning only afferent connections with $\bar{Q}^0$. Nonetheless, the weight optimization aims to find the best solution as measured by the matrix distance with respect to the objectives (Fig 7). Nonetheless, our results lay the foundation for covariance perceptrons with multiple layers, including linear feedback by recurrent connectivity in each layer. The important feature in its design is the consistency of covariance-based information from inputs to outputs, which enables the use of our covariance-based equivalent of the delta rule for error back-propagation [23]. The generalization to higher statistical orders seems a natural extension for the proposed mathematical formalism, but requires dedicated input structures and is thus left for future work.

Although our study is not the first one to train the recurrent connectivity in a supervised manner, our approach differs from previous extensions of the delta rule [21] or the back-propagation algorithm [23], such as recurrent back-propagation [26] and back-propagation through time [27–29]. Those algorithms focus on the mean activity based on first-order statistics and, even though they do take temporal information into account (related to the successive time points in the trajectories over time), they consider the inputs as statistically independent variables. Moreover, unfolding time corresponds to the adaptation of techniques for feedforward networks to recurrent networks, but it does not take the effect of the recurrent connectivity as in the steady-state dynamics considered here. We have shown that this stationary assumption is not an issue for practical applications, even though signals may strongly deviate from Gaussian distributions like the MNIST dataset. Further study about finding the best

regularities in input signals for classification, like comparing covariances and profiles of the average trajectories for MNIST digits, is left for future work. In the context of unsupervised learning, several rules were proposed to extract information from the spatial correlations of inputs [42] or their temporal variance [43]. Because the classification of time-warped patterns can be based on the second-order statistics of the input signals [43], we foresee a potential application of our supervised learning scheme, as the time-warping transformation preserves the relative structure of covariances between the input signals (albeit not their absolute values).

The reduction of dimensionality of covariance patterns —from many input nodes to a few output nodes— implements an "information compression". For the same number of input-output nodes in the network, the use of covariances instead of means makes a higher-dimensional space accessible to represent input and output, which may help in finding a suitable projection for a classification problem. It is worth noting that applying a classical machine-learning algorithm, like the multinomial linear regressor [19], to the vectorized covariance matrices corresponds to $nm(m-1)/2$ weights to tune, to be compared with only $nm$ weights in our study (with $m$ inputs and $n$ outputs). We have made here a preliminary exploration of the "capacity" of our covariance perceptron by numerically evaluating its performance in a binary classification when varying the number of input patterns to learn (Fig 6). The capacity for the classical perceptron has been the subject of many theoretical studies [34, 44, 45]. For the binary classification of noiseless patterns based on a single readout, the capacity of the classical perceptron is equal to $2m$, twice as much as its number of inputs. In contrast, we have used a network with two outputs that classifies based on the covariance or the variance difference in Fig 4. A formal comparison between the capacities of the covariance and classical perceptrons has been made in a separate paper [35]. Note that a theory for the capacity in the "error regime" was also developed for the classical perceptron [46], which may be relevant here to deal with non-perfect classification and noisy time series (Figs 5 and 6).

## Learning and functional role for recurrent connectivity

Our theory shows that recurrent connections are crucial to transform information contained in time-lagged covariances into covariances without time lag (Fig 9). Simulations confirm that recurrent connections can indeed be learned successfully to perform robust binary classification in this setting. The corresponding learning equations clearly expose the necessity of training the recurrent connections. For objectives involving both covariance matrices, $\bar{Q}^0$ and $\bar{Q}^1$, there must exist an accessible mapping $(P^0, P^1) \mapsto (Q^0, Q^1)$ determined by $A$ and $B$. The use for $A$ may also bring an extra flexibility that broadens the domain of solutions or improve the stability of learning, even though this was not clearly observed so far in our simulations. A similar training of afferent and recurrent connectivity was used to decorrelate signals and perform blind-source separation [40]. This suggests another possible role for $A$ in the global organization of output nodes, like forming communities that are independent of each other (irrespective of the patterns).

The learning equations for $A$ in Theory for learning rules (Methods) can be seen as an extension of the optimization for recurrent connectivity recently proposed [47] for the multivariate Ornstein-Uhlenbeck (MOU) process, which is the continuous-time version of the MAR studied here. Such update rules fall in the group of natural gradient descents [48] as they take into account the non-linearity between the weights and the output covariances. We have shown that a much simpler approximation of the solution of the Lyapunov equation for the weight updates gives a quasi identical performance (Fig (8)). This approximation greatly

reduces the computational cost of the covariance-based learning rule. It is expected that it may be insufficient when the recurrent connectivity grows and results in strong network feedback, in which case Eq (35) may be expanded to incorporate higher orders in *A*.

Another positive feature of our supervised learning scheme is the stability of the recurrent weights *A* for ongoing learning, even when there is no mapping that satisfies all input-output pairings (Fig 7). This is in line with previous findings [39, 49] and in contrast with unsupervised learning like STDP that requires stabilization or regularization terms, in biology known as "homeostasis", to prevent the problematic growth of recurrent weights that often leads to an explosion of the network activity [36, 37, 50]. It also remains to be explored in more depth whether such regularizations can be functionally useful in our framework, for example to improve classification performance.

### Extensions to non-linear neuronal dynamics and continuous time

In Section the capacity has been evaluated only for the case of linear dynamics. Including a non-linearity, as used for classification with the classical perceptron [21], remains to be explored. Note that for the classical perceptron a non-linearity applied to the dynamics is in fact the same as applied to the output; this is, however, not so for the covariance perceptron. The MAR network dynamics in discrete time used here leads to a simple description for the propagation of temporally-correlated activity. Several types of non-linearities can be envisaged in recurrently connected networks of the form

$$\mathrm{d}x_i^t = \psi(x_i^t) + \phi\left(\sum_j C_{ij} x_j^t\right) + \mathrm{d}\zeta_i^t \ . \tag{19}$$

Here the local dynamics is determined by $\psi$ and interactions are transformed by the function $\phi$. Such non-linearities are expected to vastly affect the covariance mapping in general, but special cases, like the rectified linear function, preserve the validity of the derivation for the linear system in Network dynamics (Methods) in a range of parameters. Another point is that non-linearities cause a cross-talk between statistical orders, meaning that the mean of the input may strongly affect output covariances and, conversely, input covariances may affect the mean of the output. This opens the way to mixed decoding paradigms where the relevant information is distributed in both, means and covariances. Extension of the learning equations to continuous time MOU processes requires the derivation of consistency equations for the time-lagged covariances. The inputs to the process, for consistency, themselves need to have the statistics of a MOU process [51]. This is doable, but yields more complicated expressions than for the MAR process.

### Learning and (de)coding in biological spiking neuronal networks

An interesting application for the present theory is its adaptation to spiking neuronal networks. In fact, the biologically-inspired model of spike-timing-dependent plasticity (STDP) can be expressed in terms of covariances between spike trains [8, 9], which was an inspiration of the present study. The weight structure that emerges because of STDP is determined by and reflects the the spatio-temporal structure of the input spike trains [11, 52]. STDP-like learning rules were used for object recognition [53] and related to the expectation-maximization algorithm [54]. Although genuine STDP relates to unsupervised learning, extensions were developed to implement supervised learning with a focus on spike patterns [4, 49, 55–58]. A common trait of those approaches is that they mostly apply to feedforward

connectivity only, even though recently also recurrently-connected networks have been considered.

Instead of focusing on the detailed timing in spike trains in output, our supervised approach could be transposed to shape the input-output mapping between spike-time covariances, which are an intermediate description between the full probability distribution of spike patterns (too complex) and firing rate (too simple). As such, our approach allows for some flexibility concerning the spike timing (e.g. jittering) and characterization of input-output patterns, as was explored before for STDP [11]. An important implications of basing information on covariance-based patterns is that they do not require a reference start time, because the coding is embedded in relative time lags. The robustness of such representations in spiking activity is also compatible with the large variability of spiking activity observed in experiments. This contrasts with supervised learning schemes of spike trains with detailed timing that have attracted a lot of recent interest [4, 49, 55]. Our theory thus opens a novel and promising perspective to learn temporal structure of spike trains and provides a theoretical ground to genuinely investigate learning in recurrently connected neuronal networks.

In addition to the computational approximation of the covariance-based learning rule, another key question for biological plausibility is whether our scheme can be implemented in a local rule, meaning that the weight updates should be calculated from quantities available by the pre- and post-synaptic neurons. Moreover, the empirical covariances should ideally be computed online. In the learning equations such as Eq (11), the matrix $U^{ik}P^0B^T$ involved in the update of weight $B_{ik}$ can be reinterpreted as a product of input and output, since its matrix element indexed by $(i', j')$ is simply $(U^{ik}P^0B^T)_{i'j'} = \delta_{i'i}\langle x_k^t(Bx^t)_{j'}^T\rangle = \delta_{i'i}\langle x_k^t y_{j'}^t\rangle$ after using $y^t = Bx^t$ according to Eq (5) with $A = 0$. Such average quantities can be obtained in an online manner by smoothing the product of activities $x_k^t y_j^t$ over several time steps. In the presence of recurrent connectivity, the learning rule for a given connection can be approximated to use only information from "parent" neurons that connect to the target neuron of the tuned connection, as detailed in Eq (18). Although this leads to a decrease in training performance (Fig (8)), the output covariance pattern can still be shaped toward an objective. One can also note that there is no reason for separating afferent and recurrent connections in the biology, hence the calculations in Eq (18) that ignore afferent connections. It remains to further explore how to approximate more efficiently the covariance-based learning rule in a local manner in the network.

Here we have used arbitrary covariances for the definition of input patterns, but they could be made closer to examples observed in spiking data, as was proposed earlier for probabilistic representations of the environment [14]. It is important noting that the observed activity structure in data (i.e. covariances) can not only be related to neuronal representations, but also to computations that can be learned (here classification). Studies of noise correlation, which is akin to the variability of spike counts (i.e. mean firing activity), showed that variability is not always a hindrance for decoding [30, 31]. Our study instead makes active use of activity variability and is in line with recent results about stimulus-dependent correlations observed in data [59]. It thus moves variability into a central position in the quest to understand biological neuronal information processing.

## Methods

Example Python scripts to reproduce some key figures are available at https://github.com/MatthieuGilson/covariance_perceptron.

## Network dynamics

Here we recapitulate well-known calculations [32] that describe the statistics of the activity in discrete time in a MAR process in Eq (5), which we recall here:

$$y_i^t = \sum_j A_{ij} y_j^{t-1} + \sum_k B_{ik} x_k^t \ . \tag{20}$$

Our focus are the self-consistency equations when the multivariate outputs $y_i^t$ are driven by the multivariate inputs $x_k^t$, whose activity is characterized by the 0-lag covariances $P^0$ and 1-lag covariances $P^1 = (P^{-1})^{\mathrm{T}}$, where T denotes the matrix transpose. We assume stationary statistics (over the observation period) and require that the recurrent connectivity matrix $A$ has eigenvalues in the unit circle (modulus strictly smaller than 1) to ensure stability. To keep the calculations as simple as possible, we make the additional hypothesis that $P^{\pm n} = 0$ for $n \geq 2$, meaning that the memory of $x_k^t$ only concerns one time lag. Therefore, the following calculations are only approximations of the general case for $x_k^t$, which is discussed in the main text about Fig 9. Note that this approximation is reasonable when the lagged covariances $P^n$ decrease exponentially with the time lag $n$, as is the case when inputs are a MAR process.

Under those conditions, we define $R_{ik}^\tau = \langle y_i^{t+\tau} x_k^t \rangle$ and express these matrices in terms of the inputs as a preliminary step. They obey

$$R^\tau = A R^{\tau-1} + B P^\tau \ . \tag{21}$$

Because we assume $P^{\pm n} = 0$ for $n \geq 2$, we have the following expressions

$$\begin{aligned}
R^{-n} &= 0 \quad \text{for } n \geq 2 \ , \\
R^{-1} &= B P^{-1} \ , \\
R^0 &= A B P^{-1} + B P^0 \ .
\end{aligned} \tag{22}$$

Using the expression for $R$, we see that the general expression for the zero-lagged covariance of $y_i^t$ depends on both zero-lagged and lagged covariances of $x_k^t$:

$$\begin{aligned}
Q^0 &= A Q^0 A^{\mathrm{T}} + B P^0 B^{\mathrm{T}} + A R^{-1} B^{\mathrm{T}} + B R^{-1\mathrm{T}} A^{\mathrm{T}} \\
&= A Q^0 A^{\mathrm{T}} + B P^0 B^{\mathrm{T}} + A B P^{-1} B^{\mathrm{T}} + B P^{-1\mathrm{T}} B^{\mathrm{T}} A^{\mathrm{T}} \ .
\end{aligned} \tag{23}$$

The usual (or simplest) Lyapunov equation [32] in discrete time corresponds to $P^{-1} = P^{1\mathrm{T}} = 0$ and the afferent connectivity matrix $B$ being the identity with $n = m$ independent inputs that are each sent to a single output. Likewise, we obtain the lagged covariance for $y_i^t$:

$$\begin{aligned}
Q^1 &= A Q^1 A^{\mathrm{T}} + B P^1 B^{\mathrm{T}} + A R^0 B^{\mathrm{T}} + B R^{-2\mathrm{T}} A^{\mathrm{T}} \\
&= A Q^1 A^{\mathrm{T}} + B P^1 B^{\mathrm{T}} + A B P^0 B^{\mathrm{T}} + A A B P^{-1} B^{\mathrm{T}} \ .
\end{aligned} \tag{24}$$

Note that the latter equation is not symmetric because of our assumption of ignoring $P^{\pm n} = 0$ for $n \geq 2$.

## Theory for learning rules

We now look into the gradient descent to reduce the error $E^\tau$, defined for $\tau \in \{0, 1\}$, between the network covariance $Q^\tau$ and the desired covariance $\bar{Q}^\tau$, which we take here as the matrix

distance:

$$E^\tau = \frac{1}{2} \parallel Q^\tau - \bar{Q}^\tau \parallel^2 \equiv \frac{1}{2} \sum_{i_1, i_2} (Q^\tau_{i_1 i_2} - \bar{Q}^\tau_{i_1 i_2})^2 \ . \tag{25}$$

The following calculations assume the tuning of $B$ or $A$, or both.

Starting with afferent weights, the derivation of their updates $\Delta B_{ik}$ to reduce the error $E^\tau$ at each optimization step is based on the usual chain rule, here adapted to the case of covariances:

$$\Delta B_{ik} = -\eta_B \frac{\partial E^\tau}{\partial B_{ik}} = -\eta_B \sum_{i_1, i_2} \frac{\partial E^\tau}{\partial Q^\tau_{i_1 i_2}} \frac{\partial Q^\tau_{i_1 i_2}}{\partial B_{ik}} = -\eta_B \frac{\partial E^\tau}{\partial Q^\tau} \odot \frac{\partial Q^\tau}{\partial B_{ik}} \ , \tag{26}$$

where $\eta_B$ is the learning rate for the afferent connectivity and the symbol $\odot$ defined in Eq (11) corresponds to the sum after the element-wise product of the two matrices. Note that we use distinct indices for $B$ and $Q^\tau$. Once again, this expression implies the sum over all indices $(i', j')$ of the covariance matrix $Q^\tau$. The first terms $\frac{\partial E^\tau}{\partial Q^\tau_{i_1 i_2}}$ can be seen as an $n \times n$ matrix with indices $(i_1, i_2)$:

$$\frac{\partial E^\tau}{\partial Q^\tau} = Q^\tau - \bar{Q}^\tau \ . \tag{27}$$

The second terms in Eq (26) correspond to a tensor with 4 indices, but we now show that it can be obtained from the above consistency equations in a compact manner. Fixing $j$ and $k$ and using Eq (23), the "derivative" of $Q^0$ with respect to $B$ can be expressed as

$$\begin{aligned}
\frac{\partial Q^0}{\partial B_{ik}} &= A \frac{\partial Q^0}{\partial B_{ik}} A + \frac{\partial B}{\partial B_{ik}} P^0 B^\mathrm{T} + B P^0 \frac{\partial B}{\partial B_{ik}}^\mathrm{T} + A \frac{\partial B}{\partial B_{ik}} P^{-1} B^\mathrm{T} + A B P^{-1} \frac{\partial B}{\partial B_{ik}}^\mathrm{T} \\
&\quad + \frac{\partial B}{\partial B_{ik}} P^{-1\mathrm{T}} B^\mathrm{T} A^\mathrm{T} + B P^{-1\mathrm{T}} \frac{\partial B}{\partial B_{ik}}^\mathrm{T} A^\mathrm{T} \ .
\end{aligned} \tag{28}$$

Note that the first term on the right-hand side of Eq (23) does not involve $B$, so it vanishes. Each of the other terms in Eq (23) involves $B$ twice, so they each give two terms in the above expression —as when deriving a product. The trick lies in seeing that

$$\frac{\partial B_{i'k'}}{\partial B_{ik}} = \delta_{i'i} \delta_{k'k} \tag{29}$$

where $\delta$ denotes the Kronecker delta. In this way we can rewrite the above expression using the basis $n \times m$ matrices $U^{ik}$ that have 0 everywhere except for element $(i, k)$ that is equal to 1. It follows that the $n^2$ tensor element for each $(i, k)$ can be obtained by solving the following equation:

$$\begin{aligned}
\frac{\partial Q^0}{\partial B_{ik}} &= A \frac{\partial Q^0}{\partial B_{ik}} A + U^{ik} P^0 B^\mathrm{T} + B P^0 U^{ik\mathrm{T}} + A U^{ik} P^{-1} B^\mathrm{T} + A B P^{-1} U^{ik\mathrm{T}} \\
&\quad + U^{ik} P^{-1\mathrm{T}} B^\mathrm{T} A^\mathrm{T} + B P^{-1\mathrm{T}} U^{ik\mathrm{T}} A^\mathrm{T} \ ,
\end{aligned} \tag{30}$$

which has the form of a discrete Lyapunov equation:

$$X = AXA^\mathrm{T} + \Sigma \tag{31}$$

with the solution $X = \frac{\partial Q^0}{\partial B_{ij}}$ and $\Sigma$ being the sum of 6 terms involving matrix multiplications.

The last step to obtain the desired update for $\Delta B_{ik}$ in Eq (26) is to multiply the two $n \times n$ matrices in Eqs (30) and (27) element-by-element and sum over all pairs $(i_1, i_2)$ —or alternatively vectorize the two matrices and calculate the scalar product of the two resulting vectors.

Now turning to the case of the recurrent weights, we use the same general procedure as above. We simply substitute each occurrence of $A$ in the consistency equations by a basis matrix (as we did with $U^{ik}$ for each occurrence of $B$), once at a time in the case of matrix products as with the usual derivation. The derivative of $Q^0$ in Eq (23) with respect to $A$ gives

$$\frac{\partial Q^0}{\partial A_{ij}} = A\frac{\partial Q^0}{\partial A_{ij}}A^{\mathrm{T}} + V^{ij}Q^0 A^{\mathrm{T}} + AQ^0 V^{ij\mathrm{T}} + V^{ij}BP^{-1}B^{\mathrm{T}} + BP^{-1\mathrm{T}}B^{\mathrm{T}}V^{ij\mathrm{T}} \ , \tag{32}$$

where $V^{ij}$ is the basis $n \times n$ matrix with 0 everywhere except for $(i, j)$ that is equal to 1. This has the same form as Eq (31) and, once the solution for the discrete Lyapunov equation is calculated for each pair $(i, j)$, the same element-wise matrix multiplication can be made with Eq (27) to obtain the weight update $\Delta A_{ij}$.

Likewise, we compute from Eq (24) the following expressions to reduce the error related to $Q^1$:

$$\begin{aligned}\frac{\partial Q^1}{\partial B_{ik}} = & A\frac{\partial Q^1}{\partial B_{ik}}A + U^{ik}P^1 B^{\mathrm{T}} + BP^1 U^{ik\mathrm{T}} + AU^{ik}P^0_{k_1 k_2}B^{\mathrm{T}} + ABP^0 U^{ik\mathrm{T}} \\ & + AAU^{ik}P^{-1}B^{\mathrm{T}} + AABP^{-1}U^{ik\mathrm{T}} \ ,\end{aligned} \tag{33}$$

and

$$\begin{aligned}\frac{\partial Q^1}{\partial A_{ij}} = & A\frac{\partial Q^1}{\partial A_{ij}}A^{\mathrm{T}} + V^{ij}Q^1 A^{\mathrm{T}} + AQ^1 V^{ij\mathrm{T}} + V^{ij}BP^0 B^{\mathrm{T}} \\ & + V^{ij}ABP^{-1}B^{\mathrm{T}} + AV^{ij}BP^{-1}B^{\mathrm{T}} \ .\end{aligned} \tag{34}$$

These expressions are also discrete Lyapunov equations and can be solved as explained before.

In numerical simulation, the learning rates are fixed to 0.01.

**Computational approximation of covariance-based learning rule.** The weight updates are given by solutions of the Lyapunov Eq (31), which can be expressed in terms of power of the recurrent connectivity $A$. We consider the drastic approximation that only retains the zeroth order and ignores all powers of $A$ in the solution:

$$X = \sum_{p \geq 0} A^p \Sigma A^{p\mathrm{T}} \simeq \Sigma \ . \tag{35}$$

It follows that the weight updates thus computed are simply given by matrix products, which dramatically reduces the computational cost of their calculation. Practically, this approximation consists in discarding the terms $A\frac{\partial Q^0}{\partial B_{ik}}A^{\mathrm{T}}$, $A\frac{\partial Q^0}{\partial A_{ij}}A^{\mathrm{T}}$, $A\frac{\partial Q^1}{\partial B_{ik}}A^{\mathrm{T}}$ and $A\frac{\partial Q^1}{\partial A_{ij}}A^{\mathrm{T}}$ in Eqs (30), (32), (33) and (34), respectively. For the case where $P^0 \neq 0$, $P^1 \neq 0$ and $Q^0 \neq 0$ (while

$Q^1 = 0$), Eqs (30) and (32) simply become after using the approximation in Eq (35)

$$
\begin{aligned}
\frac{\partial Q^0}{\partial B_{ik}} &= U^{ik}P^0B^{\mathrm{T}} + BP^0U^{ik\mathrm{T}} + AU^{ik}P^{-1}B^{\mathrm{T}} + ABP^{-1}U^{ik\mathrm{T}} \\
&\quad + U^{ik}P^{-1\mathrm{T}}B^{\mathrm{T}}A^{\mathrm{T}} + BP^{-1\mathrm{T}}U^{ik\mathrm{T}}A^{\mathrm{T}} \\
\frac{\partial Q^0}{\partial A_{ij}} &= V^{ij}Q^0A^{\mathrm{T}} + AQ^0V^{ij\mathrm{T}} + V^{ij}BP^{-1}B^{\mathrm{T}} + BP^{-1\mathrm{T}}B^{\mathrm{T}}V^{ij\mathrm{T}} \; .
\end{aligned}
\tag{36}
$$

Eq (36) can thus be seen as a cut at the second order in $A$. It is worth noting that the approximation for the weight update of $A$ still involves the terms $V^{ij}Q^0A^{\mathrm{T}} + AQ^0V^{ij\mathrm{T}}$ that come from $AQ^0A^{\mathrm{T}}$ in Eq (23) following the stationarity assumption.

**Classical perceptron rule for mean patterns.** As a comparison, we now provide the equivalent calculations for the weight update for the tuning of the mean activity of the network by considering Eq (20) without recurrent connections ($A_{ij} = 0$). We thus define for each output neuron the mean activity $Y_i = \sum_t y_i^t$ in the observation window ($1 \le t \le T$). Optimizing the output mean vector $Y$ to match a desired objective $\bar{Y}$ corresponds to the linear version of the classical perceptron [19], which can be achieved relying on a gradient descent to reduce the error $E^{\mathrm{m}}$:

$$
E^{\mathrm{m}} = \frac{1}{2} \| Y - \bar{Y} \|^2 \equiv \frac{1}{2}\sum_j (Y_j - \bar{Y}_j)^2 \; .
\tag{37}
$$

We restrict our calculations to a feedforward network with only afferent connections, $B$. In this case the network dynamics simply correspond to $Y = BX$, see Eq (9) in the main text. The derivation of their updates $\Delta B_{ik}$ to reduce the error $E^{\mathrm{m}}$ at each optimization step is based on the usual chain rule:

$$
\Delta B_{ik} = -\eta_B \frac{\partial E^{\mathrm{m}}}{\partial B_{ik}} = -\eta_B \sum_j \frac{\partial E^m}{\partial Y_j}\frac{\partial Y_j}{\partial B_{ik}} = -\eta_B \frac{\partial E^m}{\partial Y_i}\frac{\partial Y_i}{\partial B_{ik}} = -\eta_B (Y_i - \bar{Y}_i)X_k \; ,
\tag{38}
$$

for the learning rate $\eta_B$. It turns out that, in the case of feedforward networks (afferent connectivity only), only the output $Y_i$ depends on $B_{ik}$ and $\frac{\partial Y_i}{\partial B_{ik}} = X_k$. This leads to the simplification above, after also using $\frac{\partial E^\tau}{\partial Y} = Y - \bar{Y}$. We obtain an update rule for $B$ that can be expressed in matrix form:

$$
\Delta B = \eta_B (\bar{Y} - Y)X^{\mathrm{T}} \; .
\tag{39}
$$

This corresponds to the (linear) mean perceptron in Fig 11C. Note that this is different from rewriting the sum over $j$ in Eq (38) as $\frac{\partial E^\tau}{\partial Y} \odot \frac{\partial Y}{\partial B_{ik}}$, with the symbol $\odot$ corresponding to the sum after the element-wise product of the two vectors here —as with matrices in Eq (11).

Now we consider a non-linear function $\phi$ as typically involved in the classical perceptron [19]:

$$
y_i^t = \phi \left( \sum_k B_{ik} x_k^t \right) \; .
\tag{40}
$$

We can tune the weights $B_{ik}$ to reduce the error in Eq (37) based on the mean activity over the observation window, but we can alternatively define an error that considers the output activities as time-dependent trajectories, that is, in a time resolved manner. In practice, we fix an observation window defined by $1 \le t \le T$ and a desired objective $\bar{y}^t$ that is a multivariate

time series, then define the error $E^{\text{ts}}$ of the output $y^t$ as the sum of vector difference between the actual and desired trajectories:

$$E^{\text{ts}} = \frac{1}{2} \parallel y^t - \bar{y}^t \parallel^2 = \frac{1}{2} \sum_j \sum_t (y_j^t - \bar{y}_j^t)^2 \ . \tag{41}$$

To take into account the non-linearity related to $\phi$, we adapt the weight update in Eq (39) since the derivative of the output with respect to the weight becomes $\frac{\partial y_i^t}{\partial B_{ik}} = \phi'(\hat{y}_i^t) \, x_k^t$ with $\hat{y}_i^t = \sum_k B_{ik} x_k^t$ being the input argument of the nonlinear function (commonly referred to as 'net'):

$$\Delta B_{ik} = \eta_B \sum_t (\bar{y}_j^t - y_i^t) \, \phi'(\hat{y}_i^t) \, x_k^t \ , \tag{42}$$

which is simply the summation of the weight updates for the corresponding errors over time. This learning rule can also be used with a constant objective $\bar{y}_j^t = \bar{y}_i$, in which case the goal of training is to tune the output mean, see the classical perceptron for time samples ('time MLR') in Fig 11C. Note that the non-linearity may capture correlations present in the inputs, as explained in the main text. In numerical simulation, we use the the logistic function for $\phi$.

The MLR corresponds to the expression in Eq (40) when ignoring the time superscript. In that case, the weights $B_{ik}$ can be trained according to Eq (42) for "static" vectors of either mean activity calculated over the observation window ('mean MLR') of the vectorized covariance matrix ('cov MLR'). In numerical simulation, we use the scikit-learn library (https://scikit-learn.org).

**Back-propagation through time (BPTT) in recurrent neural network (RNN).** Going a step further, we consider the same non-linearity applied to the recurrent dynamics in Eq (20) to build a recurrent neural network (RNN), which also typically involves readout neurons with connections from the recurrently connected neurons:

$$y_i^t = \phi\left(\sum_j A_{ij} y_j^{t-1} + \sum_k B_{ik} x_k^t\right) \ ,$$
$$z_i^t = \phi\left(\sum_j C_{ij} y_j^t\right) \ . \tag{43}$$

Following the literature [27, 28], we refer for this RNN to the neurons with activity $y_i^t$ and $z_i^t$ as hidden and output neurons, respectively. Back-propagation through time (BPTT) applies the same type of learning rule as Eq (42) to reduce the error $E^{\text{ts}} = \frac{1}{2} \sum_{j,t} (z_j^t - \bar{z}_j^t)^2$. For the "feedforward" connections $B_{ik}$ and $C_{ij}$, it simply yields:

$$\begin{aligned} \Delta C_{ij} &= \eta_C \sum_{j,t} (\bar{z}_i^t - z_i^t) \, \phi'(\hat{z}_i^t) \, y_j^t \ , \\ \Delta B_{ik} &= \eta_B \sum_{j,t} \epsilon_i^t \, \phi'(\hat{y}_i^t) \, x_k^t \ , \end{aligned} \tag{44}$$

where the error related to the activity of the hidden neuron $y_j^t$ is the back-propagation from the output error, $\epsilon_j^t = \sum_i C_{ij}(\bar{z}_i^t - z_i^t)$, and the arguments of the nonlinear functions are $\hat{y}_i^t = \sum_j A_{ij} y_j^{t-1} + \sum_k B_{ik} x_k^t$ and $\hat{z}_i^t = \sum_j C_{ij} y_j^t$, respectively. For the recurrent connections $A_{ij}$ between the hidden neurons, the weight update involves the past activity of $y_i^t$:

$$\Delta A_{ij} = \eta_A \sum_{j,t} \epsilon_i^t \, \phi'(\hat{y}_i^t) \, y_j^{t-1} \ . \tag{45}$$

Here the learning rule can be repeated a number $L$ of steps backward in time to take into account temporal effects in how the recurrent connectivity shapes the network activity:

$$
\begin{aligned}
\Delta A_{ij} &= \eta_A \sum_j \sum_{\substack{L \le t \le T \\ 0 \le u < L}} \epsilon_i^{t,u} \, \phi'(\hat{y}_i^{t-u}) \, y_j^{t-u-1} \;, \\
\epsilon_i^{t,0} &= \epsilon_i^t \;, \\
\epsilon_j^{t,u} &= \sum_i A_{ij} \epsilon_i^{t,u-1} \text{ for } 1 \le u < L \;,
\end{aligned}
\tag{46}
$$

where the error $\epsilon_i^t = \epsilon_i^{t,0}$ is back-propagated via the recurrent connectivity at each step $1 \le u < L$. Likewise, the afferent connections $B_{ik}$ are updated to reduce the error using

$$
\Delta B_{ik} = \eta_B \sum_j \sum_{\substack{L \le t \le T \\ 0 \le u < L}} \gamma_i^{t,u} \, \phi'(\hat{y}_i^{t-u}) \, x_k^{t-u} \;.
\tag{47}
$$

In numerical simulation (Fig 11C), we use the hyperbolic tangent for $\phi$ in the RNN. The learning rates are all equal $\eta_A = \eta_B = \eta_C = 0.01$ and the depth for BPTT is $L = 5$. We define the desired objective for each category as a constant time series with 1 for the output neuron corresponding to the category and to 0 for all others. Moreover, we discard the $L$ first time points that depend on initial conditions, meaning that we only use $t \in [L, T]$ in Eq (41).

## Supporting information

**S1 Appendix. Supplementary results.** Simulations based on the analytical input-output mapping for network with trained afferent and recurrent connectivities and spatio-temporal covariances.
(PDF)

## Author Contributions

**Conceptualization:** Matthieu Gilson, David Dahmen, Rubén Moreno-Bote, Andrea Insabato, Moritz Helias.

**Methodology:** Matthieu Gilson, David Dahmen, Rubén Moreno-Bote, Andrea Insabato, Moritz Helias.

**Writing – original draft:** Matthieu Gilson, David Dahmen, Rubén Moreno-Bote, Andrea Insabato, Moritz Helias.

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
