## [Decision Letter · Decision Letter 0]

15 Apr 2020

Dear Dr Gilson,

Thank you very much for submitting your manuscript "The covariance perceptron: A new paradigm for classification and processing of time series in recurrent neuronal networks" for consideration at PLOS Computational Biology.

As with all papers reviewed by the journal, your manuscript was reviewed by members of the editorial board and by several independent reviewers. In light of the reviews (below this email), we would like to invite the resubmission of a significantly-revised version that takes into account the reviewers' comments.

It is especially important that you address the following issues raised by the reviewers:

1) Reviewers 1 & 2 note some instances where the comparison to previous ML approaches is a bit impoverished. How do ML models with similar numbers of parameters perform? What do you gain beyond that which is gained by approaches that apply standard ML techniques to second-order statistics from the data? Is it really true that LSTMs or other approaches for time-series analysis cannot achieve what this model achieves?

2) Reviewer 3 was concerned about the motivation/contribution. If, as all the reviewers suspect, many of the things that this model can achieve can be achieved by existing ML techniques, then the motivation has to be a neuroscience one. But, as Reviewer 3 noted, the current neuroscience motivation is limited to some degree. Noise in the brain is potentially easily dealt with thanks to the large numbers of neurons in the brain, and STDP can be reconciled with means, as the reviewer described. Moreover, there is the issue of the biological plausibility of the weight updates that the reviewer highlighted. Arguably, the manuscript needs to do a better job of either (1) demonstrating some clear advantages over other existing ML techniques, or (2) motivating why second-order representations are an important possibility for computational neuroscientists to consider. Note: even if the learning rule is not currently biologically plausible, that's okay, but the motivation for why this should still be considered despite such a limitation has to be much stronger.

We cannot make any decision about publication until we have seen the revised manuscript and your response to the reviewers' comments. Your revised manuscript is also likely to be sent to reviewers for further evaluation.

Sincerely,

Blake A. Richards

Associate Editor

PLOS Computational Biology

Samuel Gershman

Deputy Editor

PLOS Computational Biology

Reviewer's Responses to Questions

**Comments to the Authors:**

Reviewer #1: I'd like to start my review by saying that this paper addresses some important and interesting problems. Namely, how to go beyond the typical neuroscientist and machine learning analysis of the means of time varying stimuli. Specifically, the approach here could be described as going from first to second order. It still uses an abstraction (means + covariances), but this includes at least some of the temporal information in the stimuli. This paper could be seen as a first step on the road to understanding how to use temporal information. In particular, the most interesting part of the paper for me was the suggestion in the discussion:

"In contrast to the application of a measure to time series as a preprocessing step for machine-learning algorithms like the perceptron, our scheme opens the door to a self-consistent formulation of information processing of time series in recurrent networks, where the source signal and the classifier output have the same structure."

I really like this idea of the self-consistent formulation and agree that it is essential for neuroscience to work towards this goal. With that said, the results and analysis presented here do feel like a very preliminary first step on this road only. Maybe that's necessary.

Some comments in no particular order:

One of the key claims made is about information compression and the fact that compared to using covariances as a preprocessing step in a more standard ML approach, they use many fewer parameters (nm+n^2 or O(m) where m is the number of inputs and n the number of outputs, as compared to O(m^2) if using the matrix of covariances as a preprocessing stage). This is stated explicitly at the top of page 21. This would be a very powerful point if the results were comparable. However, the ML method with more parameters has much higher performance (98% compared to 59%). It would be interesting and fairly quick to check if the results of the proposed method are better if the ML method is restricted to a similar number of parameters, for example by randomly selecting k entries in the covariance matrix with k selected so that the total number of parameters was about the same, and using MLR on only those k randomly selected entries. Would the MLR perform better or worse in this scenario?

I'd also be interested to know if the model could pick up on even higher order statistics. For example, if you designed input time series with equal means and covariances, but different higher order statistics, could it classify them or not? If so, that's a clear advantage over doing cross-correlation followed by standard ML stuff which by construction couldn't do this task. If not, then the strength of the method (purely in terms of performance) rests mainly on whether it makes more efficient use of the parameters it has.

It seems MNIST was used in the following way: the two spatial dimensions of the digits were converted into one spatial and one temporal dimension. This feels like an odd way to construct time-varying stimuli. Why not use a more natural time-varying stimuli like an auditory signal?

p16: "To our knowledge this is the first study that tunes the recurrent connectivity in a supervised manner to specifically extract temporal information when spatial information is “absent”." Really? I'd guess that LSTMs (for example) can distinguish between different temporal patterns with the same means. Maybe I'm misunderstanding something here?

p21: "It is left for future work to test whether the mean trajectories as those in Fig. 9C can be used for prediction, using classifiers for time series like long-short-term memory units." Why not just do it? I would guess this wouldn't take that long to check with a fairly standard ML toolkit?

p22: "Because our training scheme is based on the same input properties, we expect that the strengths exhibited by those learning rules also partly apply to our setting, for example the robustness for the detection of time-warped patterns as studied in [5]." I think this statement needs some more evidence to back it up or should be toned down.

I don't know the editorial policy on providing data and code for modelling studies like this. I think it would be great if the authors made their code available, as I suspect it would be rather difficult to fully reproduce their analyses from the text alone (which seems to leave some details out).

Final thing to note is that I've suggested some additional work to do in my review above. Since we are currently in the middle of a global crisis that is severely impacting on everyone's time and ability to work, I would like these to be taken only as suggestions if the authors have time and capability to do them at the moment, but that is a choice for the editors to make.

Reviewer #2: Gilson M et al., The covariance perceptron: a new paradigm for classification and processing of time series in recurrent neural networks.

The authors introduce a new theoretical framework for training linear recurrent neural networks to map multivariate input covariances into output covariances. They show that their framework can learn to classify input covariance patterns, by mapping input covariances to output nodes, whose covariance is used to identify input patterns. They extend their results to learning lag-1 temporal dynamics. They also explore the finite-sample characteristics of their approach, and the network capacity.

General comments

This manuscript presents an interesting extension of ideas developed for perceptrons, to second order statistics. The framework is quite general, and therefore it may have broad implications and utility. I found the presentation to be relatively clear and straightforward, and the authors did a good job drawing out interesting applications of their approach. They link their work to some other previous work, including using logistic regression on covariances. Perhaps a bit more development in this direction would be useful.

Specific comments

1. Although the authors develop their framework in the general context of mapping from input covariances to output covariances, some mappings would not be possible. If I understand correctly, for example, if P0 lacks the modes that one would like to see in Q0, I do not believe a B could be found that would realize the mapping. Some brief discussion of this would be useful.

2. The approach in the manuscript is developed in the context of recognizing covariance structure in the input. A fair bit of work has been done in this area, in speech recognition systems. Although these systems generally use a pre-processing step that computes some sort of Fourier or other transform on the inputs. Old approaches to this problem tended to follow the Fourier transform step with a HMM step that computed the probability of observing a given time series, under each of many HMMs that recognized different patterns, for example phonemes. I wonder if the current results can be related to some of these other approaches for doing classification on second order statistics? What are the pros and cons? Perhaps there are broader uses of the current results, beyond classification, that can be considered? Also, instead of using a pre-processing step that effectively computed the covariances, some approaches suggested learning Kalman filters, one for each category to be recognized, and then computing the probability of observing individual time-series from each Kalman filter. How would the current approach relate to those approaches?

Reviewer #3: In this paper, the authors propose what I believe is a fundamentally new approach to learning: adjust weights to achieve a target lagged covariance matrix among neurons in a population. They derive learning rules, and show that the scheme can perform reasonably well when classifying moving MNIST digits.

The work seems solid, and the idea is interesting, but it's not clear to me that it's relevant for either neuroscience or machine learning. The motivation seems to be that neurons are noisy, so classification based on means may not be a good idea. That's what I extracted from the second and third paragraphs of the introduction, but I will admit that I did not fully agree with the logic. For instance, the first sentence of the second paragraph of the introduction reads

"The representation of the stimulus identity by the mean firing activity alone is, however, challenged by two observations in biological neuronal networks."

The two observations were that synaptic plasticity depends on pre-post timing and activity in cortex is variable. But the latter doesn't argue very strongly against mean activity being a bad representation; in this day and age, noise is not hard to deal with. Spike timing dependent plasticity is somewhat more problematic for mean-based rules, although when learning rates are low it can be cast as a firing rate covariance rule (see Ocker et al., PLOS CB 11:e1004458, 2015). So it also doesn't provide a strong argument that the mean is a bad representation.

Motivation aside, one might ask about biological plausibility. As the authors point out in the discussion, this is an open question. But it seems worse than that: Methods, Eq. 26, indicates that to compute the weight updates it's necessary to solve a discrete Lyapunov equation on every timestep. It's hard to imagine how neurons could do that. In fact, the complexity scales as n^3, so for large populations even computers have problems.

The authors do show that their method can do an OK job classifying moving digits, and it does better than the mean-based models they looked at. However, I would be shocked if an RNN wasn't perfect on this task. So it's not an obvious advance over standard networks.

Overall, then, I would say it's an interesting idea, but there do not seem to be any potential applications to either neuroscience or machine learning.

Minor comment: it wasn't clear to me what rule was used on the moving digits problem. But maybe I missed it?

**Have all data underlying the figures and results presented in the manuscript been provided?**

Reviewer #1: No: As far as I can tell, no data has been provided with this manuscript, and the source code for the models has not been made available either. I couldn't tell from the journal's pages if this is a requirement for a modelling paper or not, but I would strongly encourage the authors to make this code available as the descriptions in the paper are not sufficiently detailed to reproduce the work (see comments above).

Reviewer #2: Yes

Reviewer #3: Yes

PLOS authors have the option to publish the peer review history of their article (what does this mean?). If published, this will include your full peer review and any attached files.

Reviewer #1: No

Reviewer #2: No

Reviewer #3: No
---

## [Decision Letter · Decision Letter 1]

1 Jul 2020

Dear Dr Gilson,

Thank you very much for submitting your manuscript "The covariance perceptron: A new paradigm for classification and processing of time series in recurrent neuronal networks" for consideration at PLOS Computational Biology. As with all papers reviewed by the journal, your manuscript was reviewed by members of the editorial board and by several independent reviewers. The reviewers appreciated the attention to an important topic. Based on the reviews, we are likely to accept this manuscript for publication, providing that you modify the manuscript according to the review recommendations.

Please modify the manuscript to address Reviewer 3's remaining points. In particular, please be sure to be crystal clear with readers about the extent to which the actual model used here has a non-local learning rule.

Sincerely,

Blake A. Richards

Associate Editor

PLOS Computational Biology

Samuel Gershman

Deputy Editor

PLOS Computational Biology

[LINK]

Please modify the manuscript to address Reviewer 3's remaining points. In particular, please be sure to be crystal clear with readers that the extent to which the actual model used here has a non-local learning rule.

Reviewer's Responses to Questions

**Comments to the Authors:**

Reviewer #1: My thanks to the authors for their careful response to my review.

This revision addresses all my technical points about this manuscript, although I do think that the way the part about the number of parameters compared to MLR is written (p23) is still a bit misleading. I think it would be better to come clean that it doesn't do as well as MLR even with the same number of parameters, but may have other advantages.

Even though the case made for the covariance perceptron is not overwhelming in my opinion (since it isn't obviously an improvement in terms of machine learning, and biological plausibility is unclear), I'm supportive of publishing this paper, and more of this sort of work that tries to go beyond standard rate-based learning methods.

Reviewer #2: The authors have addressed my concerns. I have no further comments.

Reviewer #3: I don't feel massively strongly about this paper, but I am still not convinced about the locality of the learning rules. Take, for instance, the local learning rule in Fig. 8. Assuming I understand things, the learning rule is (when P^{-1}=0) something like

Delta A_ij \\propto sum_kl delta Q_ik A_kl <y_l^t y_j="">.

(It would be really nice if this were written down somewhere, since it's an important equation.) If we make the rather extreme approximation that

sum_l A_kl y_l^t = y_k^(t+1},

we have

Delta A_ij \\propto sum_k delta Q_ik <y_k^(t+1) y_j="">.

The term in brackets could be computed with a running average, but that term is stored in synapse j on neuron k. Even if the sum over k is restricted to connected pairs, I still don't see how that information can be transferred to synapse j on neuron i. Without at least some sort of plan for doing that, this just does not seem relevant for neuroscience. And to make things worse, that's a very extreme approximation, so even if the above update rule could be computed locally, it's not clear that performance would be good.

If this paper is published, the authors should be _very_ clear about exactly how local the learning rule is.

Also, three minor comments.

1. The authors seem to imply that noise is a problem for rate-based approaches, but not for the approach advocated here. However, the covariance has to be estimated from data, and second order statistics are, I believe, more sensitive to noise than first order statistics. So this approach does not seem to have an advantage with respect to noise. But maybe I misinterpreted their claim?

2. I didn't really understand Fig. 3. How many different patterns are being trained? All the covariance matrices look about the same to me.

3. I found this paper really hard to read. Maybe it's fundamentally hard, but I don't think so. Unfortunately, I don't really understand things well enough to make concrete suggestions. Except to write down the learning rules as I did above; at least it's possible to look at them and figure out what has to be computed.</y_k^(t+1)></y_l^t>

**Have all data underlying the figures and results presented in the manuscript been provided?**

Reviewer #1: **No: **Data and code have not yet been made available, and should be.

Reviewer #2: Yes

Reviewer #3: Yes

PLOS authors have the option to publish the peer review history of their article (what does this mean?). If published, this will include your full peer review and any attached files.

Reviewer #1: No

Reviewer #2: **Yes: **Bruno Averbeck

Reviewer #3: No
---

## [Editor Report · Decision Letter 2]

7 Jul 2020

Dear Dr Gilson,

We are pleased to inform you that your manuscript 'The covariance perceptron: A new paradigm for classification and processing of time series in recurrent neuronal networks' has been provisionally accepted for publication in PLOS Computational Biology.

Before your manuscript can be formally accepted you will need to complete some formatting changes, which you will receive in a follow up email. A member of our team will be in touch with a set of requests. Also, please be sure to give the final version of your manuscript a very careful read for typos, etc.

Best regards,

Blake A. Richards

Associate Editor

PLOS Computational Biology

Samuel Gershman

Deputy Editor

PLOS Computational Biology

---

## [Editor Report · Acceptance letter]

2 Oct 2020

PCOMPBIOL-D-20-00256R2 

The covariance perceptron: A new paradigm for classification and processing of time series in recurrent neuronal networks

Dear Dr Gilson,

I am pleased to inform you that your manuscript has been formally accepted for publication in PLOS Computational Biology. Your manuscript is now with our production department and you will be notified of the publication date in due course.

With kind regards,

Laura Mallard
